# *Alliums* as Potential Antioxidants and Anticancer Agents

**DOI:** 10.3390/ijms25158079

**Published:** 2024-07-24

**Authors:** Kanivalan Iwar, Kingsley Ochar, Yun Am Seo, Bo-Keun Ha, Seong-Hoon Kim

**Affiliations:** 1National Agrobiodiversity Centre, National Institute of Agricultural Science, Rural Development Administration, Jeonju 54874, Republic of Korea; kani05@korea.kr (K.I.); ocharking@korea.kr (K.O.); 2Council for Scientific and Industrial Research, Plant Genetic Resources Institute, Bunso P.O. Box 7, Ghana; 3Department of Data Science, Jeju National University, Jeju 63243, Republic of Korea; seoya@jejunu.ac.kr; 4Department of Applied Plant Science, Chonnam National University, Gwangju 61186, Republic of Korea

**Keywords:** *Allium*, edible, bioactive compounds, antioxidants, anticancer

## Abstract

The genus *Allium* plants, including onions, garlic, leeks, chives, and shallots, have long been recognized for their potential health benefits, particularly in oxidative and cancer prevention. Among them, onions and garlic have been extensively studied, unveiling promising biological activities that are indicative of their potential as potent antioxidant and anticancer agents. Research has revealed a rich repository of bioactive compounds in *Allium* species, highlighting their antioxidative properties and diverse mechanisms that target cancer cells. Compounds such as allicin, flavonoids, and organosulfur compounds (OSCs) exhibit notable antioxidant and anticancer properties, affecting apoptosis induction, cell cycle arrest, and the inhibition of tumor proliferation. Moreover, their antioxidant and anti-inflammatory attributes enhance their potential in cancer therapy. Studies exploring other *Allium* species beyond onions and garlic have revealed similar biological activities, suggesting a broad spectrum of natural products that could serve as promising candidates for developing novel anticancer treatments. Understanding the multifaceted potential of *Allium* plants will pave the way for innovative strategies in oxidative and cancer treatment and prevention, offering new avenues for pharmaceutical research and dietary interventions. Therefore, in this review, we compile an extensive analysis of the diversity of various *Allium* species, emphasizing their remarkable potential as effective agents.

## 1. Introduction

*Allium* L. is the largest genus of the Amaryllidaceae family, with petaloid monocotyledons constituting an estimated 1063 species, distributed worldwide from northern temperate regions to alpine regions [1]. The Amaryllidaceae family is a notable source of bioactive phytochemicals known for their potential in developing drugs with diverse pharmacological activities. These compounds exhibit significant antioxidant, antimicrobial, and immunomodulatory effects, making them valuable in medicinal research and pharmaceutical applications. Natural products from Amaryllidaceae are traditionally utilized to treat noncommunicable and infectious human diseases [2]. This family is often referenced in phytochemical and pharmacological literature, either in the context of plants or their alkaloids [3]. The bioactive compounds from the Amaryllidaceae family exhibit a wide range of bioactivities, including antioxidant, anti-inflammatory, antimicrobial, antifungal, antiviral, antiplasmodial, anticarcinogenic, antispasmodic, antiplatelet, antiasthmatic, antithrombotic, antitumor, antihyperlipidemic, antihyperglycemic, antiarthritic, antimutagenic, and immunomodulatory properties. The *Allium* genus is characterized by its superior ovary; it is an inflorescence scapose umbellate that bears flowers on the top scape in a bracted umbel and membranous bracts, with species that have characteristic features differing in leaves, inflorescences, bulb/rhizome, taste, and odor. Nearly 30 species have been frequently used for human consumption purposes, even though fewer than half are subjected to farming, with the most important being onions, garlic, leeks, chives, and shallots [4]. Their diverse biological properties combat bacteria and fungi, proving beneficial in treating various gastrointestinal diseases. Cultivated primarily for dietary and medicinal purposes, the most widely utilized *Allium* species include *Allium cepa*, *A. hirtifolium*, *A. sativum*, *A. schoenoprasum*, and *A. tuberosum* [5,6]. Traditionally employed in medicinal practices, *Allium* plants are valued for their therapeutic benefits [7]. These plants boast a rich array of active phytoconstituents comprising amino acids and phenolic compounds, including flavonoids such as anthocyanins, fatty acids, 63 organosulfur molecules, saponins, organic acids, steroids, and vitamins. These obversions showcase the diversity of *Allium* species and provide a deeper understanding of the broad spectrum of antioxidant and anticancer attributes present within various *Allium* plants, highlighting their significance in potential therapeutic interventions as active agents.

## 2. Origin

The genus *Allium*’s center of evolution primarily stretches along the Irano-Turanian phytogeographical zone, with additional centers in the Mediterranean basin and Northwestern America. *Allium* species are widespread in the Northern Hemisphere [8,9]. Most *Allium* species are commonly found in the dry subtropics–boreal zone, and some grow wild in the sub-Arctic belt [9,10]. The origin of the onion (*A. cepa*) is believed to be in Persia and Baluchistan (Iran and Pakistan). There is also a possibility that onions are indigenous to the Palestine–India region. Onions have been cultivated for more than a thousand years and are not known to grow wild. Their distribution is widespread, including China, Europe, Japan, North and South Africa, and the Americas. *Allium* has been used for thousands of years as a flavor enhancer in foods and in traditional medicines [9].

## 3. Methodology

Extensive research was conducted using Google Scholar and PubMed, with relevant keywords including “*Allium* species”, “*Allium sativum*”, “*Garlic*”, “*Allium cepa*”, “*Onion*”, “*Allium ampeloprasum*”, “*Allium tuberosum*”, “*Allium fistulosum*”, “*Allium schoenoprasum*”, “*Antioxidants*”, and “*Anticancer*”. Peer-reviewed scientific articles were retrieved from Web of Science, Scopus, MEDLINE, ScienceDirect, SpringerLink, Wiley Online Library, Semantic Scholar, and Europe PMC. The articles were assessed for authenticity, reliability, and relevance, resulting in the selection of 134 articles for inclusion in this review. Accepted plant names and recognized synonyms for each species were referenced from Plants of the World Online (POWO) [1], International Union for Conservation of Nature (IUCN) categorization was determined using the IUCN Red List [11], and data related to gene banks and crop information was retrieved from Genesys-Plant Genetic Resources (https://www.genesys-pgr.org/ accessed on 3 January 2024).

## 4. IUCN Redlist *Allium* Species

The *Allium* genus is included in the IUCN Red List, with 151 species categorized as critically endangered (6 species), endangered (16 species), vulnerable (7 species), near threatened (11 species), least concern (53 species), and data-deficient (58 species) (Figure 1 and Table 1). The genus *Allium* is under threat in the environment due to the development of residential and commercial urban housing areas (20%); tourism and recreation areas (15%); intrusions and disturbances from human recreational activities (16%), roads, railroad transportation, and service corridors (14%); natural ecosystem modifications (12%); overexploitation of biological resources for intentional species target usages (11%); agriculture and aquaculture by small-holder farming (9%); nomadic grazing (9%); livestock farming and ranching by small-holder grazing and ranching (9%); and unknown scale recorded (8%). These threats have caused significant stress to the species and highlight the importance of their conservation [11].

## 5. *Allium* in Genebanks

*Allium* genetic resources have been assessed through genebanks (such as Genesys, EUROCISO, and FAO-WIEWS) for more than two decades, simplifying the exchange of large data collections globally [12]. The identification of gaps that need to be filled exclusively due to ongoing genetic loss is possible by evaluating the overall genetic resources held worldwide. Threats to genetic resources, including traditional cultivation loss, environmental destruction, and climate change, have led to an increase in extinction risks. Conserving genetic resources for agriculture, particularly for breeders, is of great importance. A highly diverse gene pool of crops is a vital asset (Figure 2 and Figure 3).

## 6. *Allium* as a Potent Resource

Plants are widely recognized as sources of medicines, and phytochemicals are essential for discovering anticancer agents. *Allium* contains strong antioxidants, sulfur, and abundant phenolic compounds, garnering significant attention in the food industry [13]. It has been used for several years in treating diabetes, arthritis, colds, cough, headache, hemorrhage, asthma, atherosclerosis, and inflammatory diseases [14]. In addition, it possesses antibacterial [15], antifungal, antiviral, antiprotozoal, antiproliferative, antimotility [16], and cytotoxic properties that are conventionally used for cancer treatment [17]. As the world’s second most relevant and cultivated horticulture vegetable crop, *Allium* is distributed in over 175 countries and covers approximately six million hectares of land worldwide. Approximately two-thirds (66%) of global *Allium* production emanates from Asia, with China and India being the top producers [18]. It serves as a valuable resource for researchers investigating anticancer properties. Wild *Allium* species have been used as vegetables, spices, condiments, medicine, and ornamentals, playing a significant role in hill agriculture in the Indian Himalayan region [19]. The majority of clinically approved drugs contain medicinal plants, which are common sources of phytoactive compounds with therapeutic potential and are traditionally used for treating various forms of tumors. Plant-based drugs include vincristine, camptothecin, vinblastine, topotecan, taxol, irinotecan, and podophyllotoxin. Bioactive compounds have the potential to enhance their anticancer properties [20]. Kuete et al. [21] explored 280 Korean medicinal plants from 73 families and 198 genera, finding a significant correlation between the response of cancer cell lines and the use of therapeutic plants and natural constituents for tumor treatment. Drug development can be improved by selecting and enhancing their absorption, distribution, metabolism, and excretion properties while also reducing their toxicity and side effects [22]. The practice of traditional medicine is still common in therapeutics worldwide [23].

## 7. Phytochemistry of *Allium*

*Allium* species are renowned for their abundance of secondary metabolites with notable biological activities [24]. Economically significant *Allium* crops, such as garlic and onion, play a vital role in daily diets, serving as both vegetables and medicinal ingredients. The genus *Allium* is a rich source of compounds with diverse bioactivities, both in vivo and in vitro, containing a wide variety of metabolites. Both wild and cultivated species possess numerous key metabolites and other sulfur-rich substances, such as allicin [25]. Compounds derived from *Allium* species offer a wide range of health benefits, including antiviral, antibacterial, antifungal, antidiabetic, anticancer, antiplatelet, antispasmodic, antiseptic, anthelmintic, antithrombotic, antiasthmatic, carminative, antioxidant, anti-inflammatory, antihypertensive, hypoglycemic, hypotensive, lithontriptic, and hypocholesterolemic properties [26].

Alkaloids have been identified from this genus, including three pyridine-N-oxide alkaloids with disulfide functional groups isolated from *A. stipitatum*: 2-(methyldithio) pyridine-N-oxide, 2-(methylthiomethyl)dithio] pyridine-N-oxide, and 2,2′-dithio-bis-pyridine-N-oxide. Additionally, the thiosulfinate natural product allicin, commonly found in the genus *Allium*, was synthesized from 2-(methyldithio) pyridine-N-oxide using a straightforward method by Kitson and Loomes and isolated as yellow and orange oils [25].

Flavonoids and their derivatives have been isolated from numerous *Allium* species, including *A. cepa*, *A. sativum*, and *A. schoenoprasum*. These flavonoids exhibit antioxidant, antitumor, anti-inflammatory, and antimutagenic activities [27]. Various types of flavonoids, such as flavone, flavonol, and flavanone, have been identified across different *Allium* species.

*Allium* species contain high amounts of carbohydrates, with significant carbohydrate levels also found in *A. cepa* and *A. porrum*. *A. cepa* contains fructo-oligosaccharides, while *A. porrum* contains active polysaccharides like glucuronic acid, galactose, and rhamnose. These polysaccharides, separated by water extraction, have high polyuronic and protein content but low sugar levels. *A. sativum* primarily contains reducing sugars such as glucose, fructose, sucrose, and maltose, which are known to stimulate the human immune system [28].

Sulfur compounds such as di-allyl sulfide, sulfinate, allyl propyl sulfide, and S-methyl-L-cysteine sulfoxide have been isolated from *Allium* species. Allicin (diallyl thiosulfinate), a defense molecule found in *A. sativum*, exhibits a broad range of biological activities, including cancer, diabetes, and cardiovascular disease prevention [27]. Cysteine sulfoxides (alliin) have been isolated in several *Allium* species, including *A. sativum*. Twenty-seven volatile sulfur-rich compounds, including sulfides; disulfides; trisulfides; and tetrasulfides with ethyl, butyl, and pentyl groups, have been extracted from *A. tuberosum* [27,29]. Thioethyl and thiopentyl compounds have also been reported in *A. schoenoprasum*. These sulfur-containing compounds exhibit potent antioxidant and antitumor activities.

## 8. Antioxidant Properties of Various *Allium* Species

Antioxidants play a crucial role in protecting our bodies from oxidative stress caused by free radicals and reactive oxygen species (ROS). This oxidative stress occurs due to an imbalance between the formation of ROS and their detoxification, leading to cellular damage. Chronic oxidative stress can result in several diseases, including cancer, coronary heart disease, and osteoporosis [30]. Free radicals, such as superoxide anion (O_2_•), perhydroxyl radicals (HO_2_•), hydroxyl radicals (•OH), and nitric oxide, along with other species like hydrogen peroxide (H_2_O_2_), singlet oxygen (^1^O_2_), hypochlorous acid (HOCl), and peroxynitrite (ONOO-), can attack biomolecules, especially the polyunsaturated fatty acids in cell membranes [31]. The formation of ROS begins with the uptake of oxygen (O_2_), which activates NADPH oxidase, producing superoxide anion radicals. These radicals are then converted to hydrogen peroxide (H_2_O_2_) by superoxide dismutase (SOD) [32]. Antioxidants interrupt the chain reactions of free radicals by donating electrons to stabilize them without becoming free radicals themselves [33].

Antioxidants are classified into two types based on their activity: enzymatic and non-enzymatic. Enzymatic antioxidants include enzymes such as glutathione peroxidase (GPx), catalase (CAT), and superoxide dismutase (SOD), which catalyze the neutralization of free radicals and ROS [34]. Non-enzymatic antioxidants, found in natural materials such as fruits and onions, include flavonoids, alkaloids, carotenoids, and phenolic groups [31,34]. The antioxidant activity of these compounds can be assessed using various methods, including the DPPH free radical-scavenging assay, oxygen radical absorbance capacity (ORAC) assay, Trolox equivalent antioxidant capacity (TEAC) assay, ferric-reducing antioxidant power (FRAP) assay, cupric reducing antioxidant capacity (CUPRAC) assay, and the reducing power assay. *Allium* species play a significant role in human health as they contain various health-promoting bioactive compounds, such as flavonoids (polyphenols), sulfur compounds, vitamins, and minerals with antioxidant activity [35]. In this comprehensive review, we have extensively explored the antioxidant potential exhibited by diverse *Allium* species shown in Table 2.

### 8.1. Allium ampeloprasum L.

*A. ampeloprasum* (var. *kurrat*) originates from West Asia and S. Europe and can be cultivated in various regions. It is a good source of fiber, zinc, polyunsaturated fatty acids, and linoleic acid, as well as lipophilic and hydrophilic bioactive compounds with antioxidant activity [36]. Analysis of its extracts, including butanol, methanol, and ethyl acetate fractions, has revealed the existence of the total flavonoid content (TFC), total phenol content (TPC), caffeic acid, coumaric acid, chlorogenic acid, gallic acid, kaempferol, quercetin, tannin, and vanillic acid, all known for their high antioxidant activities [37]. Italian landraces onion’s volatile compounds and sulfur-containing compounds exhibit high antioxidant activity, with high pungency (9–14 μmol/g FW) due to alliinase activity and high levels of citric and malic acids in all parts. Fructose is the most abundant soluble sugar, followed by sucrose and glucose, with the growth temperature affecting the quality parameters [38]. Organosulfur compounds in garlic, elephant garlic, and onion have been evaluated, revealing significant *γ*-glutamyl peptides and high alliin in elephant garlic but not in onion. Antioxidant abilities measured using ABTS, DPPH, and ORAC radical-scavenging activity showed that garlic had higher activity compared to elephant garlic and onion. This indicates a connection between organosulfur compounds and antioxidants in *Allium* species [39].

### 8.2. Allium cepa L.

*A. cepa* compared with *A. fistulosum* spring cultivars plant parts were studied with the catalase, glutathione-peroxidase, superoxide-dismutase, and peroxidase activities with autumn *A. fistulosum*. *A. fistulosum* exhibits higher SOD, C-ase, GP-ase activities, while *A. cepa* L. has high P-ase and SOD activity in leaves. *A. fistulosum* spring has a high resistance to oxidative stress [40]. *A. fistulosum* (green-leaf and white-sheath var.) antioxidant activity was high in red plants, and that of the yellow variety was equal to green onion and lower in white Welsh onion when studied using FRAP and TEAC. Similarly, TFC was high in red and lower in yellow, green, and white Welsh onion. *A. cepa* contains quercetin (a major flavonoid), while *A. fistulosum* has kaempferol. Green Welsh onion is an effective antioxidant, similar to yellow onion, and it is a source of kaempferol. The thermal treatment of green Welsh onion results in an increase in antioxidant activity but a decrease in flavonoid content compared to yellow and red varieties of *A. cepa* [41].

*A. cepa* varieties (green, red, violet, and white) have been studied for their TPCs (gallic, ferulic, quercetin, protocatechuic acid, and kaempferol), which range widely from 4.6 to 74.1 mg g^−1^ GAE. In addition, antioxidant (13.6–84.1%) and free-radical-scavenging activities (0.1–15.2 mg ml^−1^ in IC_50_), as well as the concentration of efficient (4.3–660.8 mg mg^−1^ in EC_50_) and antiradical power (0.15–23.2), have been reported. Dried outer layers of red and violet varieties inhibit lipid peroxidation, non-site-specific hydroxyl radical-induced deoxyribose degradation, and nitro blue tetrazolium chloride (NBT) reduction, while the inner layers exhibit a high Fe^+^-chelating capacity. The outer layers of non-utilized red varieties are rich in quercetin with high bioactivity, significantly protecting DNA damage by free radicals [42]. Indian onion cultivars N-53, Pusa red, Sel-383, and Sel-402 (red cultivars) exhibit high TPC (>100 mg/100 g), while TFC is high in Pusa red, Sel-402, H-44, and N-53 (>40 mg/100 g). Pusa white flat, Pusa white round, and early grano (white cultivars) consistently show low TFC, TPC, and antioxidant activity. The total antioxidant activity, measured through FRAP and CUPRAC, is high in N-53, Pusa red, Sel-126, and Sel-383 cultivars. Sel-383 and Pusa red are cultivars with a high antioxidant content [43].

*The A. cepa* L. var. *tropeana* (red onion) seed shows high levels of calcium, crude protein, fiber, oil, and potassium, as well as a low sodium content and six cysteine derivatives. In addition, *S*-*propyl mercapto*-*cysteine* has been recently recorded. Both FRAP and DPPH assays indicate a significant increase in antioxidant capacity after boiling, though culinary preparation causes a notable loss of cysteine derivatives in water [44]. Garlic and red onion are commonly used in Malaysian cuisine. Their contents with medicinal and antioxidant properties have been compared. The results suggest that red onion has a higher total phenolic content (53.43 ± 1.72 mg GAE 100 g^−1^) and EC_50_ value compared to garlic (37.60 ± 2.31 mg GAE 100 g^−1^), indicating stronger free radical-scavenging activity in red onion. FIC assay data show that red onion has a higher Fe^+^-chelating effect (45.00 ± 1.73%) than garlic, while both exhibit slightly higher ion-chelating effects than BHA (43.14 ± 1.07%) but lower than EDTA (97.9 ± 0.07%). The analyses also indicate a negative correlation between DPPH radical-scavenging activity, FIC, and TPC. The evaluation of *Allium* species as natural antioxidant sources is crucial for future prospects [45]. Methanolic extracts of *A. cepa* L. (green, red, and white subspecies), *A. kurrat* L., *A. porrum* L., and *A. sativum* L. have revealed that *A. cepa* (sub sp. red) and *A. porrum* exhibit high phenolic contents and high in vitro antioxidant activity, as assessed through DPPH radical, phosphomolybdate, and reducing power assays [46].

### 8.3. Allium porrum L. and Allium roseum var. Odoratissimum (Desf.) Coss

*A. porrum* ethanolic extracts have a TPC of 93.34 mg/g and a TFC of 25.14 mg/g. They also contain apigenin, quercetin, rosmarinic acid, and rutin, which exhibit antioxidant and DPPH radical-scavenging properties and are associated with TPC and TFC. *A. roseum* var. *odoratissimum* methanolic extracts have been found to have high TFC and TPC in the flowers and leaves, as well as apigenin, kaempferol derivatives, and luteolin, which have high antioxidant activity. Extract analyses show that *Enterococcus faecium* is more liable, and the stalk extract inactivates *Bacillus subtilis, Enterococcus faecium*, *E. coli*, and *Staphylococcus aureus*, indicating antibacterial activity [47].

### 8.4. Allium sativum L.

*A. sativum* L. is well-known for its antioxidant activity due to the presence of unstable and irritating organo-sulfur contents. When garlic is extracted for an extended period, it produces an odorless aged garlic extract that contains stable and water-soluble organo-sulfur compounds that can scavenge free radicals and prevent oxidative damage. It is important to use caution when heating *Alliums* and treating them with acid during food preparation and processing to preserve their antioxidant activity [48]. The formation of allicin in bulbs has been analyzed in bakeri garlic, Chinese chive, Chinese leek, garlic, onion, shallot, and scallion to evaluate their antioxidant activity using a liposome model. In addition, pH treatments and salt concentration have been reported to have no effect on the antioxidant activity of these foods. However, the antioxidant activity of most foods is affected by heat (65/100 °C) and a low pH [48].

Garlic contains high levels of sulfur and polyphenol compounds with antioxidant activity (AA). The shelf lives of fresh garlic and commercial products have been analyzed in a study. The findings indicate that fried garlic exhibits high antioxidant activity, while the free radical-scavenging activity declines over time in commercial products due to a decrease in TPC [49]. The red onion peel (*A. cepa* L.) ethyl acetate (EA) fraction exhibits a high TFC (165.2 ± 3.2 mg QE g^−1^), TPC (384.7 ± 5.0 mg GAE g^−1^), ARP (75.3 ± 4.5), AOA (97.4 ± 7.6%), and RP (1.6 ± 0.3 ASE mL^−1^). It also contains ferulic, gallic, kaempferol, quercetin, and protocatechuic acids, which have a high antioxidant capacity against hydroxyl radicals, ferric chloride-induced lipid peroxidation, nitric oxide, protein fragmentation, and superoxide anion radicals. It also inhibits the *Salmonella typhimurium* strains (TA102) via tobacco-induced mutagenicity, and plasmid pUC18 DNA hydroxyl radical-induced nicking as antimutagenic activity could be used as a natural antioxidant [50].

The *A. sativum* polysaccharide, consisting of fructose, glucose, and galactose (monosaccharides), with a *β*-glycosidic bond, has a potent scavenging capacity for superoxide anions and hydroxyl radicals [51]. Onion cultivars have been evaluated for different physicochemical properties, including TBARS (malondialdehyde), TPC, and Trolox equivalent antioxidant capacity. The results indicate variations in carbohydrate contents that affect the essence and suitability of the onions. The outermost layers of the bulb show a high concentration of antioxidant compounds, with a slight decline in the innermost layers [52].

Garlic dry extract, which contains flavonoid aglycones and phenols, has been evaluated using the DPPH method. It reduces DPPH radical formation, demonstrating free radical-scavenging activity, neutralizes hydrogen peroxide, and inhibits lipid peroxidation (LP) in Fe^2+^/ascorbate and Fe^2+^/H_2_O_2_ induction systems [53]. The black, fresh, and steamed garlic extracts in hot water and ethanol show differences in flavonoid content, with higher levels in the ethanol extract compared to the hot water extract. Phenolic compounds are stable and have been studied to determine their antioxidant capacities. Hydroxyl radical and DPPH radical-scavenging activity was high in the ethanolic extract, with over 60% inhibition (conc. 5 mg mL^−1^). Black garlic shows high reducing power and antioxidant activity for linoleic acid, particularly after 4 days of storage [54].

Garlic and onion (green, yellow, red, and purple) have been examined for their chain-breaking activity, H_2_O_2_ scavenging, free radical-scavenging (RAS), reducing capacity, and TPCs. Garlic has a high RAS, while green onion shows a low RAS. Green onion shows a higher (0.48) chain-breaking activity than the garlic extract (0.029). The high ability of onion and garlic to scavenge hydrogen peroxide (60–90%) is well known. Garlic also possesses a high reducing capacity (196%), indicating its potency [55].

In vitro investigations of the antioxidant activity of hydroethanolic extracts (15%) from leaves and bulbs of various endemic Italian Alliums, including *A. neapolitanum* Cyr., *A. roseum* L., and *A. subhirsutum* L., using the DPPH test and FRAP tests indicate that the leaves have high antioxidant activity [56]. The evaluation of TPC and antioxidant properties of *A. cepa* L. (white, red, and yellow), *A. sativum* L., *A. schoenoprasum* L., and *A. ursinum* L. using the Folin–Ciocalteu reagent DPPH revealed an antioxidant activity that ranged from 12.29% to 76.57%. In addition, chives possess a high TPC and antioxidant activity, while white onions have the least [57].

*A. atroviolaceum*, *A. dictyoprosum*, *A. nevsehirense*, *A. scrodoprosum subsp. rotundum*, and *A. sivasicum* extracts have been assessed using *β*-carotene/linoleic acid and DPPH free radical-scavenging tests. *A. atroviolaceum* (IC_50_ of 79.0 ± 2.75 μg mL^−1^) shows the highest activity, while the inhibition of linoleic acid oxidation due to *A. atroviolaceum* and *A. dictyoprosum* are similar (71.2% ± 2.20% and 72.3% ± 1.20%). The synthetic antioxidant BHT exhibited 96% inhibition [58]. Overall, *A. sativum* is a promising antioxidant agent.

### 8.5. Allium subhirsutum L. and Allium paradoxum (M.Bieb.) G.Don

*A. subhirsutum* L. extracts contain p-coumaric acid as a major TPC, along with amino acids, amino alcohols, alkaloids, amides, fatty amides, glycerolipids, limonoids, octadecanoids, aminoglycoside antibiotics, sterol lipids, cationic sphingoids, isoprenoids, fatty acyls, octadecanoids, and lipids. In vitro assays using DPPH indicate antioxidant activity through antiradical activity and reducing power. These extracts inhibit the growth of Mat-LyLu and Walker 256/B cells, induce apoptosis, and exhibit strong chemo-preventive properties by reducing angiogenesis and osteolytic metastases [59]

The aerial part extract of *A. paradoxum* has higher TFC and TPC than the bulb extract. It also exhibits higher antioxidant activity and reducing power compared to the bulb, but it is not as effective as vitamin C. The DPPH radical-scavenging activity (890.9 ± 43.2 and 984.9 ± 33.5 µg/mL in IC_50_) and iron (Fe^2+^)-chelating ability (530 ± 24 and 959 ± 47 µg/mL in IC_50_) are higher in the aerial parts compared to the bulbs. Both extracts show hydrogen peroxide-scavenging activity. It exhibits antioxidant and antihemolytic activity [60].

### 8.6. Allium Species

*A. atroviolaceum* Bois., *A. flavum* L., *A. scorodoprasum* L., *A. sphaerocephalum* L., *A. vienale* L., and cultivars of *A. cepa* L., *A. nutans* L., *A. fistulosum* L., *A. pskemense* B. Fedtsch, *A. sativum* L., *A. schenoprasum* L., and *A. vienale* L. leaves have been quantified for chlorophylls (a, b), carotenoids, soluble proteins, TFCs, Vit-C, and hydroxyl radicals; malonyldialdehyde superoxide; and reduced glutathione. They have also been evaluated for antioxidative activities using catalase, glutathione peroxidase, peroxidase, and superoxide dismutase enzymes [61]. Štajner et al. also analyzed different plant parts of *A. nutants* L. [62], *A. psekemense* B. Fedtsch [63], *A. schoenoprasum* L. [64], *A. giganteum* [65], *A. ursinum* [66], and *A. sphaerocephalon* [67]. ESR analysis with phosphate buffer (pH 7) extract indicates that DMPO-OH spin adducts are reduced by 78.48% (*A. nutants*) and 54.3% (*A. psekemense*), while PBN-OH radical adducts are reduced by 64.51% (*A. sphaerocephalon*), 74.19% (*A. giganteum*), and 87.61% (*A. ursinum*).

Lee et al. [68] utilized wild edible plants as a resource for the creation of new crops and analyzed their antioxidant and anti-obesity properties. Korean wild vegetables *A. tuberosum* Rottl., *A. senescens* L., *A. thunbergii* G. Don., and *A. sacculiferum* Maxim. have been evaluated for their antioxidant activity using ABTS+ and DPPH-scavenging methods, as well as their anti-adipogenic effects on adipocytes. This study indicates that *A. tuberosum* and *A. sacculiferum* have a large amount of phenol, and they also contain caffeic acid, a compound that protects against adipocytes and has anti-adipogenic properties. *A. thunbergii*, *A. tuberosum,* and *A. sacculiferum* exhibit the highest levels of antioxidative and antiadipogenic effects, with potential benefits against obesity [68].

*A. tenuissimum* flowers exhibit superior antioxidant activity and inhibitory effects on nitrosation [69]. Volatile components have been identified through GC-MS analysis, with terpenoid compounds and sulfurous compounds, specifically dimethyl trisulfide, found in the highest concentrations. The antibacterial effectiveness against foodborne pathogens and the antioxidant impact have also been evaluated through the scavenging capacity of DPPH, ABTS+, and OH. Antimicrobial activity is present in the oil and aids in protection against *Aspergillus flavus*, *Bacillus subtilis*, *E. coli*, *Saccharomyces cerevisiae*, and *Staphylococcus aureus*, serving as an antibacterial and antioxidant agent [70].
ijms-25-08079-t002_Table 2Table 2Antioxidant activities of different *Allium* species.*Allium* SpeciesPlant PartsExtractsConstituentsAssayReferences*A. ampeloprasum*Leavesand SeedsMethanol, ethanol, hexane, petroleum ether, chloroform, and deionized waterAscorbic acid, dehydroascorbic acid, oxalic acid, glutamic acid, malic acid, citric acid, succinic acid, total *α*-Tocopherol, *β*-Tocopherol, *γ*-Tocopherol, *δ*-Tocopherol; gallic acid, ellagic acid, caffeic, coumaric, tannic, vanillic, chlorogenic, rutin and quercetin, 3-caffeoylquinic acid1,1-diphenyl-2-picrylhydrazyl (DPPH) radical-scavenging activity, reducing power, inhibition of *β*-carotene bleaching, thiobarbituric acid reactive substances (TBARS), ferric-reducing antioxidant power (FRAP) assay[36,37] *A. cepa*BulbsAqueous, methanol, ethanolChlorogenic acid, gallic acid, ferulic acid, kaempferol, quercetin, Propionaldehyde, 2-Methyl-2-pentenal, Furfuraldehyde, 5-Methyl-2-Furfuraldehyde, 1-Propanethiol, Propylene sulfide, Dimethyl sulfide, Methyl propyl disulfide, cis-Methyl-1-propenyl disulfide, 5-Methyl-1,3-thiazole, trans-Methyl-1-propenyl disulfide, 3,4-Dimethyl thiophene, Methyl-2-propenyl disulfide, Dipropyl disulfide, 1,2,4-Trithiolane, trans-Propenyl propyl disulfide, cis-Propenyl propyl disulfide, Methyl propyl trisulfide, Dipropyl trisulfide, 1,2-Cyclopentanedione, Butyrolactone, Furfuryl alcohol, malic, citric, tartaric, oxalic, ascorbic, succinic, and pyruvic acid; ferulic, gallic and protocatechuic acid, quercetin, kaempferol, MalondialdehydeDPPH and free radical-scavenging activities (FRSA)CUPRAC, DPPH, FRAP; antioxidant activity (AOA) *β*-carotene and linoleic acid, antimutagenic activity; Trolox equivalent antioxidant capacity (TEAC), thiobarbituric acid (TBA) assay; ferrous ion-chelating assays; phosphomolybdate and reducing power assays[38,42,43,45,46,52,55]*A. sativum*, *A. ampeloprasum*, and *A. cepa*Bulbs-Alliin, allicin, cycloalliin, isoalliin, methiinOxygen radical absorbance capacity (ORAC) values and 2,2-diphenyl-1-picrylhydrazyl (DPPH) and 2,2′-azinobis(3-ethylbenzothiazoline-6-sulphonic acid) (ABTS) radical-scavenging activities[39]*A. cepa* and *A. fistulosum*Leaves, stalks, and rootsAqueous-Catalase (C-ase), glutathione-peroxidase(GP-ase) peroxidase (P-ase), and superoxide-dismutase (SOD)[40]*A. fistulosum*BulbsAqueous-Trolox equivalent antioxidant capacity (TEAC) and ferric-reducing antioxidant power (FRAP) assays[41]*A. cepa* var. *tropicana*SeedsHeat-treated by boiling waterAlanine, arginine, asparagine, aspartic acid, Gaba, glutamic acid, glutamine, glycine, hystidine, proline, serine, threonine, tyrosine, and valineFerric-reducing/antioxidant power, DPPH[44]*A. kurrat*BulbsMethanolFerulic, gallic, and protocatechuic acid; quercetin, kaempferol; steroids; terpenoids; and saponinsDPPH radical, phosphomolybdate, and reducing power assays[46]*A. porrum*Stem and leavesMethanol, Ethanol and AqueousApigenin, chlorogenic acid, ferulic acid, gallic acid, dihydroxybenzoic acid, caffeic acid, kaempferol glucoside, myricetin, naringenin, quercetin glucoside, protocatechuic acid, quercetin, rosmarinic acid, rutin, sinapenic acid, syringic acid, and vanillic acidDPPH radical, phosphomolybdate and reducing power assays, and Minimum Inhibitory Concentration (MIC)[46,71] *A. roseum* var. *odoratissimum*Leaves, flowers, stalks, and bulbsMethanolApigenine, kaempferol-3-*O*-glucoside, kaempferol-3-*O*-beta-*D*-glucoside-7-*O*-alpha-*L*-rhamnoside, kaempferol 3,7-di-*O*-rhamnoside, kaempferol-3-Glucuronide, and luteolineDPPH, 2-deoxyribose, ferric-reducing antioxidant power (FRAP), reducing power assay [47]*A. sativum*, *A. bakeri*, *A. odorum*, *A. tuberosum*, *A. fistulosum*, *A. cepa*, and *A. ascalonicum*BulbsAqueousAllicinThiobarbituric acid (TBA)[48]*A. sativum*BulbsMethanol, aether-petroleum ether and aqueous, aqueous–ethanolMethanolic extract; crude polysaccharide; ferulic acid, gallic acid, kaempferol, protocatchuic acid, quercetinDPPH (1,1-diphenyl-2-picrylhydrazyl) assay, *β*-carotene/linoleic acid assay, and Rancimat method; scavenging activity of superoxide anions, free radical-scavenging capacity (RSC), hydrogen peroxide and hydroxyl radical-scavenging activity; ferrous ion-chelating assays; phosphomolybdate and reducing power assays[45,46,49,51,53,54,55] *A. neapolitanum*, *A. subhirsutum*, *A. roseum*Leaves, flowers bulblets, and flowers, bulbsAqueous ethanolGallic acid (TPC)DPPH and FRAP assay[56]*A. cepa*, *A. sativum*, *A. schoenoprasum*, *A. ursinum*.BulbsEthanolGallic acid (TPC)Antioxidant activity (AOA), DPPH assay[57]*A. atroviolaceum*, *A. dictyoprosum*, *A. nevsehirense*, *A. sivasicum*, *A. scrodoprosum* subsp. *rotundum*Whole PlantMethanol-DPPH free radical-scavenging and *β*-carotene/linoleic acid assays[58]*A. subhirsutum*Whole PlantEthanolTPC, TFC, 2-methylene-5-(2,5dioxotetrahydrofuran-3-yl)-6-oxo--10,10-dimethylbicyclo [7:2:0] undecane; (22S)-1*α*,22,25-trihydroxy-26,27-dimethyl-23,23,24,24-tetradehydro-24ahomovitaminD3/(22S)-1al; L-4-Hydroxy-3-methoxy-amethylphenylalanine; 1-nonadecanoyl-2-(5Z,8Z,11Z,14Z,17Zeicosapentaenoyl)-sn-glycerol; TG(16:1(9Z)/17:2(9Z,12Z)/20: 5(5Z,8Z,11Z,14Z,17Z))[iso6]; 11*α*-acetoxykhivorin; Methyl gamboginate; C16 Sphinganine; 4-Oxomytiloxanthin; Sebacic acid; Linolenoyl lysolecithin; 3*β*, 7*α*, 12*α*-Trihydroxy-5*α*-cholestan- 26-oic acid; N-(2-hydroxyethyl) stearamide; Cepharanthine; 6*α*-Hydroxy Castasterone; 6-DeoxocastasteroneDPPH, reducing power, Malignant MatLyLu and Walker 256/B Cell Lines Culture, 3-(4,5-Dimethylthiazol-2-yl)-2,5-diphenyltetrazolium bromide (MTT), Hoechst 33,342 (apoptosis) Assay, AS Extract on breast cancer skeletal metastases[59]*A. paradoxum*Leaves and BulbsAqueous-methanolGallic acid (TPC), quercetin (TFC)DPPH, reducing power, nitric oxide and hydrogen peroxide scavenging, metal-chelating, antihemolytic activities[60]*A. flavum*, *A. sphaerocephalum*, *A. atroviolaceum*, *A. vienale*, *A. scorodoprasum*, *A. nutans*, *A. fistulosum*, *A. vienale*, *A. pskemense*, *A. schenoprasum*, *A. cepa*, *A. sativum*LeavesCrude with 1 mol/L K_2_HPO_4_Gallic acid (TPC), quercetin (TFC), reduced glutathione, vitamin C, and soluble proteinsSuperoxide dismutase, catalase, peroxidase, glutathione peroxidase, quantities of malonyldialdehyde superoxide, and hydroxyl radical-scavenging activities[61]*A. nutans*Leaves, bulb, and rootCrude with 1 mol/L K_2_HPO_4_Gallic acid (TPC), quercetin (TFC), reduced glutathione, vitamin C, and soluble proteinsSuperoxide dismutase, catalase, peroxidase, glutathione peroxidase, quantities of malonyldialdehyde superoxide, and hydroxyl radical-scavenging activities[62]*A. psekemense*Leaves, stalk, and bulbCrude with 1 mol/L K_2_HPO_4_Gallic acid (TPC), quercetin (TFC), reduced glutathione, vitamin C, and soluble proteinsSuperoxide dismutase, catalase, peroxidase, glutathione peroxidase, quantities of malonyldialdehyde superoxide, and hydroxyl radical-scavenging activities[63]*A. schoenoprasum*Leaves, stalk, and bulbCrude with 1 mol/L K_2_HPO_4_Gallic acid (TPC), quercetin (TFC), reduced glutathione, vitamin C, and soluble proteinsSuperoxide dismutase, catalase, peroxidase, glutathione peroxidase, quantities of malonyldialdehyde superoxide, and hydroxyl radical-scavenging activities[64]*A. giganteum*Leaves, stalk, and bulbCrude with 1 mol/L K_2_HPO_4_Gallic acid (TPC), quercetin (TFC), reduced glutathione, vitamin C, and soluble proteinsSuperoxide dismutase, catalase, peroxidase, glutathione peroxidase, quantities of malonyldialdehyde superoxide, and hydroxyl radical-scavenging activities[65]*A. ursinum*Leaves, stalk, and bulbCrude with 1 mol/L K_2_HPO_4_Gallic acid (TPC), quercetin (TFC), reduced glutathione, vitamin C, and soluble proteinsSuperoxide dismutase, catalase, peroxidase, glutathione peroxidase, quantities of malonyldialdehyde superoxide, and hydroxyl radical-scavenging activities[66]*A. sphaerocephalon*Leaves, stalk, and bulb; flowersCrude with 1 mol/L K_2_HPO_4,_
Gallic acid (TPC), quercetin (TFC), reduced glutathione, vitamin C, and soluble proteins*α*-cadinol, *β*-caryophyllene, *δ*-cadinene, 3,5-diethyl-1,2,4-trithiolane, ShyobunolSuperoxide dismutase, catalase, peroxidase, glutathione peroxidase, quantities of malonyldialdehyde superoxide, and hydroxyl radical-scavenging activities; total antioxidant capacity determined using the Phosphomolybdenum method and antimicrobial activity[67,72]*A. tuberosum*, *A. senescens*, *A. thunbergii*, and *A. sacculiferum*SeedlingsCrudeCaffeic acid, chlorogenic acid, cinnamic acid, coumaric acid, ferulic acid, gentisic acid, hesperiin, homogentisic acid, naringenin, propionic acid, protocatechinic acid, quercetin, and veratric acidABTS+ and DPPH-scavenging assays[68]*A. tenuissimum*FlowersAqueous, ethanol, ethyl acetate, and petroleum etherGallic acid (TPC), quercetin (TFC)DPPH, ABTS+, and total reducing power[69,70]*A. kurrat*Whole PlantMethanolIsorhamntin-*O*-hexoside-pentoside, Quercetin-tri-*O*-hexoside, Kaempferol-tri-*O*-hexoside, Kaempferol-tri-*O*-hexoside isomer, Kaempferol-di-*O*-hexoside, Kaempferol-*O*-trihexoside-hexuronoide, Kaempferol-di-*O*-hexoside isomer, Quercetin-*O*-hexoside, Kaempferol-di-*O*-hexoside isomer, Kaempferol-*O*-hexoside, Kaempferol-*O*-hexuronoide, Kaempferol-*O*-trihexoside-hexuronoide isomer, Kaempferol-*O*-dihexoside-hexuronoide, Acacetin-7-*O*-malonoyl hexosideDPPH, ABTS, and total antioxidant capacity; human hepatocellular carcinoma (HepG2); and human colon carcinoma (Caco-2) using neutral red assay[73]*A. astrosanguineum*Aerial partsCrude in methanolAqueous and methanolic extractsDPPH radical-scavenging assay and antimicrobial activity[74]


## 9. Anticancer and Related Activities of *Allium* Species

Throughout history, the *Allium* species, including chives, garlic, scallions, onions, and leeks, have been widely distributed and utilized in human diets worldwide. They are not only popular as food but also as herbal medicines, and a large number of species are grown as ornamental plants [75,76]. *Allium* species with unknown value should be conserved for future studies. Various studies have shown that *Allium* plants exhibit antitumor effects by inhibiting tumor proliferation both in vitro and in vivo [77,78]. Carcinogens in the environment may contribute to human cancer. *Allium* components have been reported to have anticarcinogenic activity by modulating specific and non-specific antitumor immunity. *Allium* species with bioactive compounds and anticancer activities are listed in Table 3.

### 9.1. Allium ampeloprasum L. and Allium ascalonicum L.

*A. ampeloprasum* (*A. kurrat*) contains flavonoids and phenols that play a vital role in regulating antioxidants, potentially reducing the risk of endometrial cancer and colorectal tumor recurrence. Studies indicate that both the Caco-2 and HepG2 cell lines displayed antitumor and cytotoxic effects [73]. *A. ascalonicum* L. (shallot) is a species originating from the Eastern Mediterranean and Arabian Peninsula, containing valuable sources of sulfur compounds, flavones, quercetin, isorhamnetin, glycosides, polyphenols, and furostanol saponins [79]. Mohammadi-Motlagh et al. [80] employed the aqueous extract method to assess the impact of extracts from *A. ascalonicum* on K562, Wehi164, Jurkat, and normal cell lines. Their results revealed the potential of *A. ascalonicum* in combating cancer and angiogenesis. The in vitro antiproliferative activity of the shallots ethanolic extract against HepG2 cell lines was also investigated [81], revealing the potential of shallots for use in robust anticancer treatment. Identified 3,4′-diglucoside, isorhamnetin derivatives, and quercetin derivatives exhibit activities against Hela cells and HepG2, demonstrating stronger anticancer, anti-inflammatory, and antioxidant effects of *A. ascalonicum*, and can be used for treating inflammatory and cancer-related diseases [82].

### 9.2. Allium affine Ledeb and Allium atropurpureum Waldst. & Kit

*Allium affine* Ledeb, commonly known as the Persian onion, is a bulbous perennial native of Central Asia. It produces small, pale pink to purple flowers in clusters on tall stems and is often cultivated for its ornamental value. *A. atropurpureum* Waldst, also called the Dark Purple Allium, is native to regions from Hungary to North-western Türkiye. This species typically features large, globe-shaped purple flowers on long stalks and is cultivated for its striking appearance in gardens and landscapes. *A. affine* has been demonstrated to induce a dose-dependent decrease in succinate dehydrogenase activity in OVCAR-3 cells, accompanied by growth-inhibition and cytotoxic effects [17].

Studies have revealed that *A. atropurpureum* extracts contain steroidal glycosides, furostanol glycosides, pseudofurostanol glycosides, spirostanol glycosides, and spirostanol isolates. Using a modified MTT assay, the cytotoxic effects of steroidal glycosides isolated from *A. atropurpureum* have been evaluated for their cytotoxic effects on SBC-3 human small-cell lung cancer cells. The results indicate that the steroidal glycosides display moderate cytotoxicity toward SBC-3 cells, with IC_50_ values in the range of 1.4–6.0 µM, suggesting their potential effectiveness against human small-cell lung cancer cells [83].

### 9.3. Allium atroviolaceum Boiss and A. austroiranicum R. M. Fritsch

*A. atroviolaceum* Boiss is native to the Middle East and Iran and is characterized by its distinctive deep violet flowers and slender stems. This perennial plant thrives in rocky or mountainous terrain, often found in arid or semi-arid regions. The striking floral display and adaptability of this *Allium* make it a sought-after species for ornamental purposes in xeriscaping and rock gardens. The therapeutic properties of extracts from *A. atroviolaceum* Boiss as a natural compound in traditional medicine have been demonstrated. Methanol extract from this *Allium* species has been tested for cytotoxicity against HeLa, HepG2, MCF7, and MDA-MB-231 cell lines, revealing a significant antitumor effect through apoptosis [84,85,86]. *A. austroiranicum* is a native species from Western and Southwestern Iran. The cytotoxic properties of *A. austroiranicum* extracts on HeLa, OVCAR-3, and HUVEC cells have been determined [87] and reported to be enriched in protodioscin saponins, which show high cytotoxicity against cancerous cells.

### 9.4. Allium autumnale P.H. Davis and Allium willeanum Holmboe

*Allium autumnale* and *A. willeanum* are both endemic species of the Cyprus islands. These unique endemic species represent a distinct botanical wealth of Cyprus, showcasing specific compounds that contribute to their ecological significance and potential applications in various fields such as medicine or ecology. Extracts from *A. autumnale,* including 1,2-benzenedicarboxylic acid, ethyl oleate, hexadecanoic acid, l-isoleucine, octadecamethyl-cyclononasiloxane, and tetrapentacosane, are known to possess anticancer properties [88].

*A. willeanum* contains octadecanoic acid 2-hydroxy-1-(hydroxymethyl) ethyl ester, hexadecanoic acid, 1,2-benzenedicarboxylic acid, diethyl ester, and pentadecanoic acid, which have high anticancer activities. A significant reduction in mitochondria-dependent metabolic activity and cell motility has been detected in MDA-MB-213 and MCF-7 cells. Quantitative cytotoxicity assays using trypan blue and LDH have shown a reduction in cancer cell viability, and differential activation of caspases has indicated apoptotic activity, suggesting that *A. willeanum* can be utilized in anticancer research [89]. The anticancer potential has been tested in MCF-7 and MDA-MB-231 using ethanolic extracts of these plants. The growth of melanoma is inhibited by both species, with MDA-MB-231 cells displaying faster apoptosis activity than MCF-7 cells and a notable decrease in cell motility, suggesting their potential to be effective in anticancer studies [88,89].

### 9.5. Allium cepa L.

*Allium cepa*, a native species of Central Asia, is widely distributed. It is commonly utilized in the food development industry as a source of nutritional and aromatic resources, containing abundant anthocyanin pigments, carotenoids, flavonoids, glutathione, glycosides/aglycones, organosulfur compounds, phenolics, quercetin derivatives, saponins, and tannins [12,90]. Fatty acid synthase inhibition has many inhibitory effects on cancerous cell and adipocyte proliferation [91], and its chemoprotective properties make it beneficial for decreasing the risk of gastrocancer and gastrointestinal tract cancers [92]. Veiga et al. [93] have investigated the cytotoxicity of the adrenocortical carcinoma cell line (H295R) and discovered that flavonoids could be useful in combating adrenocortical carcinoma. An ethanolic extract of onion has shown cytotoxic effects on HeLa cell lines, as assessed using the MTT assay, with an IC_50_ of 900.88 µg/mL [92]. As an adjuvant therapy for cancer, it is commonly used and has the potential to have anticancer properties.

### 9.6. Allium bakhtiaricum Regel and Allium fistulosum L.

*Allium bakhtiaricum* Regel and *Allium fistulosum* L. are distinct species within the *Allium* genus. *A. bakhtiaricum*, commonly known as Persian shallot, is a perennial herbaceous plant native to Iran, characterized by its small bulbs and pink to purple flowers. Vafaee et al. [94] have found that MDA-MB-231 cells respond favorably to chloroformic and ethyl acetate extracts of *A. bakhtiaricum* after 72 h, with significant G2/M cell cycle disruption. This finding indicates that chloroform extract (1 mg/kg/day) effectively stops the progression of mammary tumors and lowers the number of proliferative cells in the respiratory tissues, suggesting an anti-metastatic effect of the compound [94].

*A. fistulosum* L., native to China, is a species with potential medicinal properties and widely cultivated as a culinary herb throughout Asia [95]. Reports have shown that cinnamic acid derivatives, flavonoids, furostanol saponins, and thiolane-type sulfide can be effective in fighting various diseases such as colds, flu, abdominal pain, cephalgia, and arthritis, which are among the illnesses that have shown immense potential in boosting immunity against COVID-19 [96]. Welsh onion is rich in 4-hydroxybenzoic acid, allicin, gentisic acid, alliin, chlorogenic acid, ferulic acid, isoquercitrin, kaempferol, quercetin, rutin, and p-coumaric acid. The cytotoxic activity has been found to be highly sensitive to SK-MES-1 in vitro [97]. Additionally, *A. fistulosum* is capable of reducing the growth of MDA-MB-453 cell lines through caspase-mediated cell death. The anti-inflammatory and anticancer activities can be inhibited in ethanol and methanolic extracts from the non-developed bulbs of spring onion (*A. fistulosum*) [98].

### 9.7. Allium chinense G. Don

The native range of *A. chinense* spans from Southern China to Hainan. *A. chinense* has versatile compounds such as 2-methyloctacosane, tetracontane, eicosane, 10-methyl, heneicosane, octadecyl trifluoroacetate, 1-heneicosanol, phytol, tetratetracontane, perhydrofarnesyl acetone, and heptadecane 2,6-dimethyl. The leaves and bulbs of this *Allium* crop have been demonstrated to contain high concentrations of saponins, with 375 mg/g in the leaves and 163.75 mg/g in the bulbs. *Staphylococcus aureus* and *Pseudomonas aeruginosa* can be inhibited by the DPPH antioxidant scavenging activity of *A. chinense*, and the leaf extract can also inhibit *Aspergillus niger* [99]. The saponins found in *A. chinense* exhibit cytotoxic properties that modify cell morphology; induce cell death in B16 and 4T1 cells; and inhibit proliferation, cell migration, and colony formation. Furthermore, these compounds have shown a concentration-dependent effect, effectively safeguarding the spleen and liver of C57 BL/6 mice from injuries, providing further evidence of their potential value in drug development [100].

### 9.8. Allium giganteum Regel and Allium jesdianum Boiss. & Buhse

The native range of *A. giganteum* extends from North East Iran to Central Asia. Cytotoxic activity against HeLa and MCF-7 cell lines using MTT assay significantly exhibited toxicity. Butanol extracts at a concentration of 50 μg mL^−1^ in MCF-7, and 20 or 50 μg mL^−1^ in HeLa cells induce apoptosis [101]. *A. jesdianum* is native to Iraq, Central Asia, and Afghanistan. In folk medicine, it is utilized to alleviate abdominal and renal discomfort [102]. B-CPAP and Thr.C1-PI 33 (thyroid cancer cell lines) have been evaluated for viability, apoptosis, and nitric oxide production using *A. jesdianum* hydro-alcoholic extract reported with steroidal saponins (102) (Figure 4). In this particular study, the release of lactate dehydrogenase was triggered by treating cells with differing concentrations of the extract, resulting in a reduction in Bcl-2 expression by inducing apoptotic cell death, along with increases in Bax, p53, and Caspase 3 levels [103]. The viability of GBM cells can be reduced by *A. jesdianum* extract in a manner dependent on concentration and time, leading to a reduction in the gene-expression levels of MMP-2 and MMP-9. It has a high level of antioxidants and contains numerous phenolic compounds, which could be involved in its anti-cancer properties [104].

### 9.9. Allium kurtzianum Asch. & Sint. ex Kollmann and Allium leucanthum K. Koch

*A. kurtzianum,* also known as wild garlic in Türkiye, contains (+)-catechin, (−)-epicatechin, (−)-epigallocatechin, (−)-epigallocatechin gallate, acacetin, apigenin 7-glucoside, caffeic acid, chrysin, fumaric acid, herniarin, hispidulin, hyperoside, luteolin-7-rutinoside, naringenin, nepetin, nepetin-7-glucoside, quercetin, quercitrin, rhamnocitrin, and rutin from methanolic extract. It shows effective antioxidant and strong cytotoxic activities against human prostrate (ATCC CRL-1435, PC-3), lung (ATCC CRL-185, A549), and endometrial (ATCC CRL-2923, ECC-1) cell lines in vitro [105]. The extract of *A. atrosanguineum* Schrenk also has antioxidant properties, as demonstrated via DPPH radical-scavenging analysis, and exhibits antimicrobial properties against bacteria responsible for common bacterial diseases [74]. *A. leucanthum* is a Caucasian endemic species that extends into Georgia and is used in indigenous drugs. It contains spirostanol type saponins and leucospiroside A. In vitro cytotoxicity studies on crude, spirostanol, and furostanol fractions have shown promising results against lung (A549) and colon (DLD-1) cancer cell lines (5.6–8.2 µM) as a promising agent against cancer [106].

### 9.10. Allium macrostemon Bunge and Allium ochotense Prokh. (syn.: A. victorialis var. platyphyllum (Hultén) Makino)

*A. macrostemon*, known as the wild long-stamen chive, originates from East Russia, China, Korea, and Japan and is recognized as one of Korea’s medicinal plants [21]. Steroidal saponins isolated from this plant have been tested on HepG2 cells, MCF-7, NCI-H460, and SF-268, displaying selective cytotoxicity activity against NCI-H460 and SF-268 cell lines [107]. Macrostemonoside A (Figure 5) has been found to induce apoptosis in a xenograft reactive oxygenated mouse model and inhibit colorectal cancer cells [108]. Moreover, its therapeutic effects and detailed mechanisms for treating acute myocardial ischemia have been evaluated using rats [109]. *A. ochotense* extract shows antioxidant activities, as analyzed using 3-ethylbenzothiazoline-6-sulfonic acid, 1,1-diphenyl-2-picrylhydrazine, iron reduction antioxidant strength, and malondialdehyde tests. The synergistic activity of alcohol-metabolizing enzymes, such as alcohol dehydrogenase, has also been reported. DCF-DA and MTT assays indicate that oxidative stress decreases with water and 60% ethanol extracts, leading to increased cell viability. Extracts of this plant regulate proteins associated with apoptosis, such as Bcl-2, BAX, and procaspase-3 in HepG2 cells, displaying the potential to prevent ethanol-induced cytotoxicity [110].

### 9.11. Allium porrum L. and Allium pseudojaponicum Makino

*Allium porrum*, native to Iran and widespread in Europe, Southeast Asia, and West Australian regions, is commonly referred to as elephant garlic. The methanol extract demonstrates an antiproliferative effect on HT-115 and colon cancer cells. The growth of these cells is suppressed through the mitochondria-mediated caspase-dependent apoptosis mechanism, as demonstrated by Alshammari et al. [111]. *Allium porrum* extract inhibits Hep2c (27.18 ± 0.88 µg/mL), L2OB (25.89 ± 0.67 µg/mL), and Rhabdomyosarcoma (RD) (76.95 ± 11.45 µg/mL) cells in a dose-dependent manner, showing cytotoxic activity [71]. *A. ampeloprasum* var. *porrum* is rich in entadamide-*A*-*β*-*D*-glucopyranoside, significantly inhibiting MCF-7 human breast cancer cells, a finding detailed by Zamri and Hamid [112]. Saponins derived from this species exhibit cytotoxicity toward various malignant cell lines, as reported by Fattorusso et al. [113]. Harmatha et al. [114] have identified constituents of spirostanol saponins, including aginoside, 6-deoxyaginoside, and yayoisaponin A. Notably, 6-deoxyaginoside displays the highest inhibitory effect on production in mouse peritoneal cells due to its cytotoxic properties.

*Allium pseudojaponicum* Makino is a rare species native to S. Korea and Geomundo Island. Shim [115] investigated *A. pseudojaponicum* extract for its anti-aging effects on normal human epidermal keratinocytes (NHEK) through gene-expression analysis related to skin hydration, hyaluronic acid (HA) production using HA-ELISA, cell viability, and phosphokinase array. The author also demonstrated an increase in the AQP3/HAS2 gene-expression level and HA protein production in NHEK. Overexpressing mRNA levels decreased KRT1, 10, and FLG genes and the phosphorylation of CHK2 decreased the epidermal differentiation and p53 proteins. It can be used as an efficient component for anti-aging and skin hydration products.

### 9.12. Allium sativum L.

*Allium sativum* is known for its diaphoretic, diuretic, expectorant, and stimulant properties. An ethanolic extract from a single clove of garlic has demonstrated a significant reduction in oxidative toxicity and free radicals, improving liver biomarkers, cholestasis, enzyme markers, fibrosis severity, and hepatocyte architecture, potentially serving as a preventive intervention against free radicals [116]. Kurnia et al. [90] have analyzed the bioactive compounds from the extract, including alkaloids, flavonoids, glycosides, allin, astragalin, oleanolic acid, terpenoids, saponins, steroids, triterpenes, foliogarlic disulfane A1, A2, A3, foliogarlic trisulfane A1, A2, *S*-allyl-*L*-cysteine, glutamyl-*S*-allyl-*L*-cysteine, isorhamnetin 3-*O*-*β*-*D*-glucopyranoside, isoquercitrin, and reynoutrin. Garlic diallyl sulfide, administered through oral gavage to C57BL/6J mice, has been found to prevent cancer by 74% and reduce the occurrence of colorectal adenocarcinoma [117].

The oral administration of *A. sativum* at a concentration of 50 mg/100 mL in mice resulted in a notable drop in tumor volume and mortality (*p* < 0.05) [118]. An analysis of Hep-2 (human larynx) and L929 (normal mouse fibroblast) cell lines treated using an aqueous extract of garlic has indicated stronger inhibition effects after the 2nd and 3rd days. Additionally, the MTT assay in this study has revealed a significant decrease in cell viability, which is commonly used for anticancer purposes [119]. In vitro studies on the HeLa cell line demonstrate the effectiveness of aqueous garlic extract as an anticancer agent [120]. Gold nanoparticles extracted from garlic leaves exhibit notable anticancer activity against various cell lines and show potential as a novel chemotherapeutic material [121]. Furthermore, an aqueous extract of *A. cepa* utilized for silver nanoparticle synthesis has been reported to display antioxidant activity, inhibit cell proliferation, and induce apoptosis in colon cancer cells, suggesting promise in cancer treatment [122].

### 9.13. Allium saralicum R.M. Fritsch and Allium schoenoprasum L.

*A. saralicum* extract-containing, green-synthesized silver nanoparticles have anti-human breast cancer and cytotoxic effects. Silver nitrate, *A. saralicum*, and silver nanoparticles have been shown to have anti-human breast cancer effects. AU565 [AU-565], Hs 281, MDA-MB-231, and SK-BR-3 have been tested using MTT assays. T-cell lines have also exhibited the anticancer properties of AgNPs against the MDA-MB-231 cell line [123]. *A. schoenoprasum*, native to the Temperate Northern Hemisphere, is commonly known as chives and used as an appetizing condiment rich in vitamin C, carotin, and calcium [124]. Phenolic compounds found in its flowers show strong antiproliferation effects on HaCaT cells [125]. In addition, this species has antimicrobial and antifungal properties, as it prevents phagocytosis by lowering nitro-oxidative stress [72,126] and acts as a natural antioxidant [64]. Isolated spirostane-type glycosides and steroidal saponin compounds have been examined for their cytotoxic effects on colon cancer cells (HT-29 and HCT 116) [127].

### 9.14. Allium senescens L. and Allium subhirsutum L.

*A. senescens* is native to Siberia and Korea. The drug resistance of HepG2 cells can be reduced using the crude extract and *p*-coumaric acid from *A. senescens*, which induce mitochondrial apoptosis and suppress autophagy. The effectiveness of sorafenib against hepatocarcinoma is maintained by *A. senescens*, which counteracts cancer cell drug resistance [128]. The crude extract activates MAPK p38 phosphorylation, inducing caspase-dependent apoptosis and ROS production in Jurkat cells. It also inhibits IL-2 mRNA expression and NF-κB signaling through phorbol-12-myristate 13-acetate and phytohemagglutinin. The inhibitory effects on T-ALL cell growth are enhanced when the extract and axitinib/dovitinib are combined, suggesting its potential as an herbal therapy for T-ALL [129]. Adipogenic transcription factors are regulated by methanol extracts, which modulate ROS production associated with ROS-regulating genes and inhibit adipogenesis [130]. *A. subhirsutum* has a high p-coumaric acid content (1700 µg g^−1^ extract) with a strong antioxidant effect, and the extract inhibits the proliferation of Walker 256/B and Mat-LyLu cells (IC_50_ ≃/150 µg). Also, induced apoptosis adipogenic transcription factors are regulated by methanol extracts that modulate ROS production related to ROS regulatory genes and inhibit adipogenesis. Walker 256/B malignant cells are key to the compound’s chemopreventive action, which causes apoptosis and reduces angiogenesis and osteolytic metastasis [59].

### 9.15. Allium sivasicum Özhatay & Kollmann and Allium sphaerocephalon L.

*A. sivasicum* is a native species of Central Türkiye. An in vitro anti-proliferative analysis of breast cancer cells was conducted and revealed that caspase-7 protein TUNEL-positive cells expression and Ki-67 and antitumoral potential in breast cancer decreased [131]. *A. sphaerocephalon* is native to the Canary Islands, Europe, the Mediterranean region, and the Caucasus region. It is commonly known as the Round-Headed Leek. Its major constituents are shyobunol, *β*-caryophyllene, *α*-cadinol, 3,5-diethyl-1,2,4-trithiolane (isomer II), and *δ*-cadinene. Gallic acid and 3-*O*-methyl quercetin are its main compounds, showing inhibitory potential against acetylcholinesterase (AChE), butyrylcholinesterase (BuChE), and tyrosinase [132].

### 9.16. Allium stipitatum Regel (syn.: A. hirtifolium)

*A. stipitatum*, known as Persian shallot, is native to Southeast Türkiye, Central Asia, and Pakistan. *A. hirtifolim* has been investigated for its cell-growth-inhibition effects on HeLa, MCF-7, and L-929 cell lines using the MTT assay (IC_50_, 20 and 24 mg/L at 72 h), demonstrating cell growth inhibition at a non-toxic concentration. A turbidimetry assay has shown that inhibiting microtubule polymerization leads to tubulin-binding as a ligand; *A. hirtifolium* has been shown to lower the polymerization of MTs, which can be used as a powerful ligand for cancer treatment [133]. *A. stipitatum* oil contains 1-Butene, dimethyl tetrasulfide, 1-methylthio-(Z), methyl-methylthiomethyl-disulfide, and piperitenone oxide, with mono-sulfur compounds being the most abundant components. [134]. *A. stipitatum* leaf aqueous extract formulated and prepared as silver nanoparticles (AgNPs) show antioxidant properties and exhibit dose-dependent anti-cancer effects on colorectal cell lines (CW2, HCT116, and HT-29) [135]. Endothelial cell migration and angiogenesis are inhibited by the methylpyridine-1-ium-1-sulfonate compound in both in vitro and in vivo tests. MCF-7 and MDA-MB-231 cell lines show a substantial decrease in proliferation and an increase in the secretion level of the vascular endothelial growth factor [136]. Extract ointments promote wound healing in rats as they enhance wound contraction, decrease epithelialization time, increase hydroxyproline levels, and improve histological characteristics, providing therapeutic benefits [136].

### 9.17. Allium wallichii Kunth and Allium tuberosum Rottler ex Spreng

*A. wallichii* extract contains flavonoids, glycosides, steroids, terpenoids, and reducing sugars, which show significant antimicrobial and antioxidant activity. Its cytotoxic effects on the prostate cell line (PC3) (69.69 μg mL^−1^), breast cell line (MCF-7) (55.29 μg/mL), cervical cell line (HeLa) (46.51 μg mL^−1^), and Burkitt’s lymphoma cell line (3.817  ±  1.99 mg/mL) indicate anticancer activities [137]. *A. tuberosum*, known as garlic chive, is native to the Himalayas and China. The thiosulfinate isolates S-methyl 2-propen-1-thiosulfinate and S-methyl methanethiosulfinate show in vitro cytotoxicity against human tumor cells and in vivo anticancer activity in mice inoculated with Sarcoma-180 cancer cells, leading to an increase in lifespan. These isolates inhibit cancer cell lines through apoptosis, exhibiting antitumor potential [138], and activate the colon cancer (HT-29 cells) apoptosis pathway through caspase-dependent/independent mechanisms [139]. The MTT assay has been utilized to evaluate the cytotoxic effects of *A. tuberosum* partition layers on HeLa, HepG2, SK-N-MC, and MCF-7 cells [140]. The ethyl acetate layer shows a significant impact on SK-N-MC and HepG2 cell lines, with butanol being the most active in inducing quinone reductase (QR) in HepG2 cells. The butanol layer increases the reductase activity of HepG2 cells by 3.9 times. *A. tuberosum* ethyl ether and butanol layers may have anticarcinogenic and chemopreventive activities [141]. Apoptosis can prevent the proliferation of cancer cells (MDA-MB-453) [140]. The therapeutic effect on the melanoma B16F10 cell line in C57BL/6 mice is attributed to a decline in anaerobic metabolism and an increase in redox capacity [142]. The ability of americine to combat non-small cell lung cancer by promoting apoptosis and inhibiting cell proliferation has been demonstrated in NSCLC using EGFR, B-Raf, K-Ras, and PI3K [143].

### 9.18. Allium ursinum L.

*A. ursinum* is a wild garlic known as ramson and widely used as a spice and indigenous medicine in Europe, where it is utilized to reduce blood pressure and treat arteriosclerosis, diarrhea, and indigestion [66]. Sulfur compounds such as thymidine, adenosine, and astragalin kaempferol derivatives also exhibited moderate antimicrobial properties [144]. Stajner et al. [66] analyzed the activity and quantity of superoxide, malonyldialdehyde, reduced glutathione, and hydroxyl radicals in *A. ursinum* leaf extract and found an increase in these components through the action of glutathione peroxidase, peroxidase, catalase, and superoxide dismutase enzymes. Ramson watery extract affects the proliferation and apoptosis of atypical glandular cells (AGS) [145]. *A. ursinum* leaves and flower extract show anti-inflammatory properties, as they inhibit phagocytosis and reduce nitro-oxidative stress [146]. Other authors analyzed antioxidant activities and lactic acid bacterial (LAB) activities. Thiosulfinates and diallyl disulfide inhibit the growth of tumor cells by inducing the G2/M phase and apoptosis through caspase-3 activation and mitochondrial signal pathways, thereby conferring *A. ursinum*’s anti-cancer effects; leaves can be used as a probiotics and consumed by humans [147].
ijms-25-08079-t003_Table 3Table 3Anticancer activities of different *Allium* species.*Allium* Species.Plant PartsExtractsConstituents/ExtractsAssayReferences*A. kurrat*Whole PlantMethanolIsorhamntin-*O*-hexoside-pentoside, Quercetin-tri-*O*-hexoside, Kaempferol-tri-*O*-hexoside, Kaempferol-tri-*O*-hexoside isomer, Kaempferol-di-*O*-hexoside, Kaempferol-*O*-trihexoside-hexuronoide, Kaempferol-di-*O*-hexoside isomer, Quercetin-*O*-hexoside, Kaempferol-di-*O*-hexoside isomer, Kaempferol-*O*-hexoside, Kaempferol-*O*-hexuronoide, Kaempferol-*O*-trihexoside-hexuronoide isomer, Kaempferol-*O*-dihexoside-hexuronoide, Acacetin-7-*O*-malonoyl hexoside.DPPH, ABTS, and total antioxidant capacity; human hepatocellular carcinoma (HepG2); and human colon carcinoma (Caco-2) using neutral red assay[73]*A. ascalonicum*Bulbsn-hexane, chloroform, chloroform-methanol (9:1), methanol; aqueous, ethanol, Ascalonicoside A1, ascalonicoside A2 and ascalonicoside *B*, furost-5(6)-en-3*β*,22*α*-diol 1*β*-*O*-*β*-*d*-galactopyranosyl 26-*O*-[*α*-l-rhamnopyranosyl-(1→2)-*O*-*β*-*d*-glucopyranoside] (1a), (1b), and furost-5(6),20(22)-dien-3*β*-ol1*β*-*O*-*β*-*d*-galactopyranosyl 26-O-[*α*-l-rhamnopyranosyl-(1→2)-*O*-*β*-*d*-glucopyranoside;Ethanolic extractsquercetin 3,4′-diglucoside, isorhamnetin 3,4′-diglucoside, quercetin 3-glucoside, quercetin 4′-glucoside, isorhamnetin 4′-glucoside, quercetin aglycone, isrohamnetinIn vitro cytotoxicity, cell line antiproliferative, anti-growth, and anti-inflammatory activity;in vitro anticancer efficacy liver cancer cell line HepG2 using MTT (3-(4,5-dimethylthiazol-2-yl)-2,5-diphenyltetrazolium bromide) assay; DPPH, anti-inflammatory effect, and human cervical carcinoma (Hela) and human hepatocellular carcinoma (HepG2) cell lines—MTT assay[79,80,81,82]*A. atropurpureum*BulbsMethanol(25S)-26-[(*β*-*D*-glucopyranosyl)oxy]-2*α*,6*β*,22*α*-trihydroxy-5*α*-furostan-3*β*-yl *O*-*β*-*D*-glucopyranosyl-(1→2)-*O*-[*β*-*D*-xylopyranosyl-(1→3)]-*O*-*β*-*D*-glucopyranosyl-(1→4)-*β*-*D*-galactopyranoside; (25S)-2*α*,6*β*-dihydroxy-5*α*-spirostan-3*β*-yl *O*-*β*-*D*-glucopyranosyl-(1→2)-*O*-[*β*-*D*-xylopyranosyl-(1→3)]-*O*-*β*-*D*-glucopyranosyl-(1→4)-*β*-*D*-galactopyranoside; (25S)-2*α*,6*β*-dihydroxy-5*α*-spirostan-3*β*-yl *O*-*β*-*D*-glucopyranosyl-(1→2)-*O*-[4-*O*-(3S)-3-hydroxy-3-methylglutaryl-*β*-*D*-xylopyranosyl-(1→3)]-*O*-*β*-*D*-glucopyranosyl-(1→4)-*β*-*D*-galactopyranosideCytotoxicity assay SBC-3 human small-cell lung cancer cells[83]*A. atroviolaceum*Bulb; FlowersMethanol-Cytotoxic activity of MCF7 (human hormone-dependent breast cancer) and MDA-MB-231 (human non-hormone-dependent breast cancer cell line), HeLa (human cervical cancer), HepG2 (human hepatocellular cancer cell line), and 3T3 (mouse embryo fibroblast) cell lines; apoptosis; and normal 3T3 cell lines—MTT assay and apoptosis[84,85,86]*A. austroiranicum*FlowerHexane, chloroform, chloroform-methanol, and methanol-Anti-proliferative effects of OVCAR-3 (ovarian carcinoma), HeLa, and HUVEC (human umbilical vein endothelial) cell lines determined using cytotoxicity assay (MTT)[87]*A. autumnale*Bulb and stemEthanol9-octadecenoic acid, octadecamethylcyclononasiloxane; tetrapentacosane; l-Isoleucine; heptadecanoic acid; hexadecanoic acid; 1,2-benzenedicarboxylic acid, diethyl ester; dimethyltrisulfide; (−)-1 L–cyclohex-5-ene-1,3/2,4-tetrol; 14-.*β*.-h-pregna; pentadecanoic acid; and quinic acid; 9-octadecenoic acid; octadecamethylcyclononasiloxane; tetrapentacosane; heptadecanoic acid; hexadecanoic acid; 1,2-benzenedicarboxylic acid, diethyl ester; dimethyltrisulfide; (−)-1 L–cyclohex-5-ene-1,3/2,4-tetrol; 14-.*β*.-h-pregna; pentadecanoic acidIn vitro anti-proliferative, cytotoxic and anti-metastatic effects of MCF-7 and MDA-MB-231 (breast cancer) cell lines[88]*A. bakhtiaricum*Aerial part*n*-hexane, chloroform, ethyl acetate, and methanol-In vitro cytotoxicity assay of MDA-MB-231 and MCF-7 (human breast adenocarcinoma), HT-29 (human colorectal adenocarcinoma), HepG2 (liver hepatocellular carcinoma), 4 T1 (mouse mammary tumor), and NIH3T3 (mouse embryonic fibroblasts) cell lines and in vivo study using mice[94]*A. cepa*BulbsAqueous, ethyl acetate, ethanol, petroleum ether Quercetin; silver nano particles using aqueous extractsInhibitory effects on Mouse 3T3-L1 preadipocytes (fatty acid synthase) and MDA-MB-231 cell line apoptosis (human breast cancer)—MTT assay; cytotoxicity effects on adrenocortical carcinoma cell line (H295R and SW-13); HeLa (human cervix carcinoma) cell cytotoxicity activity; TROLOX, total antioxidant capacity (TAC) and DPPH-scavenging activity, and apoptosis in colorectal cancer (HT-29 and SW620) cell lines[90,91,92,93]*A. willeanum*BulbsEthanolOctadecanoic acid 2-hydroxy-1-(hydroxymethyl) ethyl ester; hexadecanoic acid; pentadecanoic acid; 1,2-benzenedicarboxylic acid, diethyl esterMetastatic effects of MCF-7 and MDA-MB-231 (breast cancer) cell lines, trypan blue, and lactate dehydrogenase (LDH) cytotoxicity assays[89]*A. fistulosum*, *A. sativum*BulbEthanolAlliin, Allicin, gentisic acid, chlorogenic acid, 4-hydroxybenzoic acid, rutin, Isoquercitrin, *p*-coumaric, Quercitrin, ferulic acid, quercetin and kaempferolNormal human fibroblasts (BJ cells) and keratinocytes (HaCaT) cell lines[97]*A. fistulosum*Leaves and bulbEthanol and methanolQuercetin and gallic acid, chlorogenic and *p*-coumaric acid,Anti-inflammatory and anticancer activity.[98]*A. chinense*Leaves and bulbHexane, ethanol Phytol, tetratetracontane, perhydrofarnesyl acetone, heptadecane, 2,6-dimethyl, 2-methyloctacosane, tetracontane, eicosane, 10-methyl, heneicosane, octadecyl trifluoroacetate, and 1-heneicosanol, saponinsDPPH antioxidant scavenging, antibacterial and antifungal activity; in vivo anticancer activity of B16 melanoma and 4T1 breast carcinoma cell lines[99,100]*A. giganteum*FlowerButanol, dichloromethane, chloroform-methanol (9:1).Steroidal saponinCytotoxic and pro-apoptotic effects on MCF-7 and HeLa cell lines using MTT assay[101]*A. hirtifolium*Bulb-AllicinNerve cell microtubules and cytotoxicity effect on HeLa, MCF-7, and L-929 cell lines[133]*A. jesdianum*Leaves and stemsEthanol, hydro-alcohol-In vitro anti-proliferative and cytotoxic effects, B-CPAP and Thr.C1-PI 33 cancer cell lines using MTT assay; Cytotoxic and migrastatic effect of Glioblastoma multiforme cell line (U87MG)[102,103,104]*A. kurtzianum*Aerial parts and bulbMethanolAcacetin, apigenin 7-glucoside, caffeic acid, (+)-Catechin, chrysin, (−)-Epicatechin, (−)-Epigallocatechin, (−)-Epigallocatechin gallate, fumaric acid, herniarin, hispidulin, hyperoside, Luteolin-7-rutinoside, naringenin, nepetin, Nepetin-7-glucoside, Quercetin, quercitrin, rhamnocitrin, rutinDPPH, FRAP activity (antioxidant), *α*-amylase and *α*-glucosidase inhibition assays (antidiabetic), DNA-protection activity (DNA nicking assay), cytotoxic activity on human prostate (ATCC CRL-1435, PC-3), human lung (ATCC CCL-185, A549), and human endometrial (ATCC CRL-2923, ECC-1) cancer cell lines using Viability Test using MTS[105]*A. macrostemon*Bulb; whole plantEthanol Steroidal saponins, 26-*O*-*β*-*D*-glucopyranosyl-5*α*-furost-25 (27)-ene-3*β*, 12*β*, 22, 26-tetraol-3-*O*-*β*-*D*-glucopyranosyl (1→2) [*β*-*D*-glucopyranosyl (1→3)]-*β*-*D*-glucopyranosyl (1→4)-*β*-*D*-galactopyranoside and 26-*O*-*β*-*D*-glucopyranosyl-5*β*-furost-20 (22)-25 (27)-dien-3*β*, 12*β*, 26-triol-3-*O*-*β*-*D*-glucopyranosyl (1→2)-*β*-*D*-galactopyranoside; Macrostemonoside A (steroidal saponin)In vitro cytotoxic activities on MCF-7, NCI-H460, SF-268, and HepG2 cancer cell linesIn vitro anti-proliferative and apoptosis effects on human colorectal cancer cell lines Caco2 and SW480[107,108,109]*A. ochotense*BulbAqueous and ethanol-ABTS, DPPH, FRAP, malondialdehyde (MAD) assays (antioxidant), Alcohol Dehydrogenase (ADH), aldehyde dehydrogenase (ALDH) MTT assays and apoptosis[110]*A. porrum*Whole plant; flowerChloroform, n-hexane, and methanol, ethyl acetate, butanol, chloroform-methanol–water Cyclotrisiloxane, 11,13, -Dimethyl-12-tetradecen-1-ol acetate, Hexadecanoic acid, 9, 12, 15-Octadecatrienoic acid, 9, 12-Octadecadienoic acid, Methyl ester Dodecanoic acid and 1,3,5, -Triazine,*β*-chlorogenin aglycone, spirostanol saponins, 12-ketoporrigenin and 2,12-diketoporrigenin (porrigenin C), cholestane, bidesmosides; Agigenin, Aginoside, 6-deoxyaginoside, Yayoisaponin A and (2*α*, 3*β*, 6*β*, 25R)-2,6-dihydroxyspirostan-3-yl *β*-*D*-glucopyranosyl-(1→3)-*β*-*D*-glucopranosyl-(1→2)-[*β*-*D*-xylopyranosyl-(1→3)]-*β*-*D*-glucopyranosyl]-(1→4)-*β*-*D*-galactopyranoside (Alliporin)In vitro cytotoxicity using MTT assay on HT-115 (human colon carcinoma) cell line, WEHI 164 (murine fibrosarcoma) J774 (murine monocyte/macrophage) cell lines); in vitro study using mouse and cytotoxicity effect determined using LDH (lactate dehydrogenase) assay[111,113,114]*A. cepa* and *A. ampeloprasum*Whole plantAqueous, ethanol, methanolAlicin, E-ajoene, *S*- allylmercapto-cysteine (SAMC), *S*-allylcysteine (SAC), Dipropyl disulfide, 4′-Desulphate-atractyloside, Entanamide *A*, Entadamide *A*-*β*-*D*-glucopyranoside, Glucoerucin, 2-Hydroxyxanthiside, Xanthiside, and XanthiazoneAnticancer effect on MCF-7 cells using cell viability (MTT assay)[112]*A. sativum*Bulbs; leavesAqueousDiallyl sulfide; gallic acid standard (TPC); gold nano particles using aqueous extractDPPH, superoxide radical, hydrogen peroxide-scavenging activity, and oxidative hepatotoxicity assessed using Rabbit; in vivo analysis of colorectal adeno carcinoma effects in mouse; in vivo analysis of murine transitional cell carcinoma using C3H/HeN female mice; in vitro cell lines study on laryngeal cancer cells (Hep-2) and L929 cells; in vitro cytotoxicity and anticancer assays on HeLa cells by trypan blue exclusion method; cytotoxicity effects on HUVEC (human normal cell line), HT-29, HCT 116 (colorectal carcinoma), HCT-8 [HRT-18] (ileocecal colorectal adenocarcinoma), and Ramos.2G6.4C10: Burkitt’s lymphoma using MTT assay[90,116,117,118,119,120,121,122]*A. saralicum*PlantAqueousSilver nano particles using aqueous extractsIn vitro cytotoxicity effect on breast cancer cell lines (SK-BR-3, MDA-MB-231, AU565 [AU-565], and Hs 281) using MTT Assay[123]*A. schoenoprasum*Flowers; whole plantMethanol, methanol-aqueous Caffeic acid, catechin, cinnamic acid, coumaric acid, ferulic acid, gallic acid, resveratrol, rutin, vanillic acid, quercetin and sinapic acid; Spirostane-type glycosides: (20*S*,25*S*)-spirost-5-en-3*β*,12*β*,21-triol 3-*O*-*α*-*L*-rhamnopyranosyl-(1→2)-*β*-*D*-glucopyranoside (**1**), (20*S*,25*S*)-spirost-5-en-3*β*,11*α*,21-triol 3-*O*-*α*-*L*-rhamnopyranosyl-(1→2)-*β*-*D*-glucopyranoside (**2**), laxogenin 3-*O*-*α*-*L*-rhamnopyranosyl-(1→2)-[*β*-*D*-glucopyranosyl-(1→4)]-*β*-*D*-glucopyranoside (**3**), and (25*R*)-5*α*-spirostan-3*β*,11*α*-diol 3-*O*-*β*-*D*-glucopyranosyl-(1→3)-[*β*-*D*-glucopyranosyl-(1→4)]-*β*-*D*-galactopyranoside. Prosapogenin A, deltonin and deltosideAntiproliferative activity on HaCaT cells using the MTT assay; in vitro cytotoxic assay on HCT 116 and HT-29 (human colon cancer) cell lines[124,125,126,127]*A. senescens*Leaves and stemsMethanol*p*-coumaric acidIn vitro cytotoxicity effect on Sorafenib-ResistantHuman HCC cells (HepG2); proliferative effect on human T-cell acute lymphocytic leukemia cells; DPPH radical-scavenging activity, MTT and NBT Assays[128,129,130]*A. sivasicum*Whole plantAqueous-In vitro cytotoxicity and apoptosis effect on breast cancer (MCF-7, MDA-MB-468) cells using (MTT) proliferation assay and in vivo study in albino Wistar rats[131]*A. stipitatum*LeavesAqueousSilver nano particles using aqueous extractsCytotoxicity of cerium oxide nanoparticles on colorectal carcinoma cells (HT-29, HCT116, and CW2)[133,134,135,136]*A. tuberosum*Whole Plant and LeavesDichloromethane, methanol, hexane, ethyl-ether, ethyl-acetate, butanol and aqueous, acetone, petrol-eum etherCrude thiosulfinates, *S*-methyl methanthiosulfinate and *S*-methyl 2-propene-1-thiosulfinate; thiosulfinates; methanol, hexane, ethylether, ethylacetate, butanol, andaqueous extract; americine, 9-Hydroxy-9,11,15-octadecatrienoic acid (9-HOTE), Di-n-octyl phthalate, 8-hydroxyoctadeca-9,12-dienoic acid (8S-HODE), 9-hydroperoxy-octadeca-10,12,15-trienoic acid [9(S)-HpOTrE], *ɑ*-Linolenic Acid, Fumaric acid-di-(2-decyl) ester, 3-ketostearic acid, 1,2-Benzene-di-carboxylic acid butyloctyl ester, *N*,*N′*-Pentamethylene-*bis*-[*s*-3-aminopropyl thiosulfuric acid, Ethyl 2E,4Z-hexadecadienoate, 9,12-Octadecadien-1-Ol, Eicosanoic acid, methyl ester, petroselinic acid, 10,16-dihydroxy-palmitic acid, Nonadecane,9-methyl, Methaphenilene, 10,11-Epoxy-3,7,11-trimethyl-2E,6E-tridecadienoic acid, Glycerol1,2-diacetate, Leucyl-glutamateIn vitro cytotoxicity on human cancer cells and in vivo apoptosis in MCF-7 cancer cells; in vitro cytotoxicity and apoptosis effect on HT-29 human colon cancer cells; HepG2, HeLa, and SK-N-MC cells using the MTT assay; in vitro effect on malignant melanoma in C57BL/6 mice; inhibitory activity against B-Raf, EGFR, K-Ras, and PI3K of non-small-cell lung cancer targets[138,139,140,141,142,143]*A.tuberosum*, *A. macrostemon*, *A. thumbergii*Whole PlantAqueous-In vitro anti-adipogenic, anti-inflammatory activities, and inhibition effect of MDA-MB-453 cancer cell proliferation[141]*A. ursinum*Whole plantAqueous, methanol-acetic acid-Proliferation and apoptosis effect of human gastric cancer cells; antioxidant and antiproliferative activity and in vitro gastrointestinal digestion on the cytotoxic activity on human malignant cell lines[144,145,146,147]*A. wallichii*LeavesAqueous ethanol
Alkaloids, coumarin, flavonoids, glycosides, quinone, reducing sugars, saponins, steroids, tannins, terpenoids
DPPH free radical-scavenging assay, anti-microbial activity, and MTT/cytotoxicity assay against B-lymphoma cancer cell lines[137]


## 10. Antioxidant and Anticancer Effects of *Allium*

*Allium* species, including *A. sativum*, *A. cepa*, and *A. porrum*, are well-known for their rich phytochemical profiles, which exhibit potent antioxidant and anticancer properties. The primary bioactive compounds found in these species are organosulfur compounds (OSCs) such as allicin, diallyl-disulfide (DADS), diallyl trisulfide (DATS), and S-allylcysteine (SAC), as well as flavonoids and saponins.

### 10.1. Antioxidant Mechanism

*Allium* phytochemicals combat oxidative stress through multiple mechanisms, including the direct neutralization of free radicals and enhancement of the body’s antioxidant defenses. Allicin, the major component in garlic, undergoes rapid transformation into various OSCs, which are responsible for its biological activity [148]. These OSCs act as electron donors to reactive oxygen species (ROS), converting them into less reactive molecules and preventing cellular damage [27]. One significant antioxidant mechanism involves the modulation of the body’s endogenous antioxidant enzyme systems. Enzymes such as catalase, glutathione peroxidase, and superoxide dismutase are crucial for detoxifying ROS. Studies have shown that DADS and DATS can upregulate the expression of these enzymes, thereby enhancing the detoxification of ROS and reducing oxidative stress [149]. Additionally, flavonoids present in *Allium* species have been found to scavenge free radicals directly and chelate metal ions that catalyze free radical formation [150]. Moreover, *Allium* phytochemicals can modulate cellular signaling pathways associated with oxidative stress. For instance, SAC has been shown to activate the nuclear factor erythroid 2-related factor 2 (Nrf2) pathway, leading to the upregulation of various antioxidant genes. This pathway plays a critical role in maintaining cellular redox homeostasis and protecting cells from oxidative damage [151].

### 10.2. Anticancer Mechanism

The anticancer effects of *Allium* phytochemicals are multifaceted, involving the induction of apoptosis, inhibition of cell proliferation, suppression of angiogenesis, and modulation of immune responses [152].

*Induction of Apoptosis:* apoptosis is a critical mechanism through which the body eliminates cancer cells. OSCs present in *A. sativum*, such as DADS and DATS, have been found to induce apoptosis in cancer cells through the activation of both intrinsic and extrinsic apoptotic pathways [153]. These compounds increase the expression of pro-apoptotic proteins (Bax, Bak) and decrease the expression of anti-apoptotic proteins (Bcl-2, Bcl-xL), leading to mitochondrial dysfunction and the release of cytochrome c. This release activates caspases, which are essential for the execution of apoptosis [154]. In addition, DATS has been shown to induce endoplasmic reticulum (ER) stress in cancer cells, which triggers the unfolded protein response (UPR) and leads to apoptotic cell death. This mechanism highlights the ability of *Allium* phytochemicals to target multiple cellular pathways to induce apoptosis in cancer cells [155].

*Inhibition of Cell Proliferation: Allium* phytochemicals inhibit cancer cell proliferation by arresting the cell cycle at various checkpoints. For instance, SAC and DADS can cause cell cycle arrest at the G2/M phase by downregulating cyclin B1 and Cdc25C and upregulating p21 and p27, which are cyclin-dependent kinase inhibitors [156,157]. This arrest prevents the progression of the cell cycle, thereby inhibiting the proliferation of cancer cells. Moreover, allicin has been shown to inhibit the growth of colon cancer cells by downregulating the expression of genes involved in cell proliferation and survival, such as c-Myc and cyclin D1. This inhibition of proliferative signaling pathways underscores the potential of *Allium* phytochemicals in cancer therapy [150].

*Suppression of Angiogenesis:* Angiogenesis, the formation of new blood vessels, is essential for tumor growth and metastasis. *Allium* phytochemicals, particularly DATS, have been shown to suppress angiogenesis by downregulating vascular endothelial growth factor (VEGF) and its receptor (VEGFR) [158]. This inhibition hinders the angiogenic signaling pathways, thereby reducing the supply of nutrients and oxygen to the tumor and inhibiting its growth. In addition, garlic-derived OSCs have been found to inhibit the migration and invasion of endothelial cells, which are crucial steps in the angiogenesis process. By targeting multiple aspects of angiogenesis, *Allium* phytochemicals can effectively suppress tumor growth and metastasis [159].

*Modulation of Immune Responses:* the immune system plays a vital role in identifying and eliminating cancer cells. *Allium* phytochemicals enhance the immune system’s ability to fight cancer by modulating the activity of various immune cells. For instance, allicin and its derivatives have been shown to boost the activity of natural killer (NK) cells and macrophages, which are essential for identifying and destroying cancer cells [160]. Also, *A. sativum* extracts have been found to enhance the production of cytokines, such as interleukin-2 (IL-2) and interferon-gamma (IFN-*γ*), which are critical for the activation and proliferation of immune cells. This immunomodulatory effect of *Allium* phytochemicals highlights their potential as adjuvants in cancer immunotherapy [157,161].

## 11. Conclusions

*Allium* species are highly desirable and widely used in culinary arts, including several wild species used in traditional and folklore medicine in certain regions. The consumption of *Allium* species reduces the risk of gastrointestinal tract cancer. Organosulfur compounds found in these species inhibit carcinogen activation; enhance immune function; and have antioxidant, antibacterial, antifungal, antiviral, and anticancer properties.

By analyzing the structure–activity relationships, we can understand the mechanisms through which diverse bioactive compounds contribute to the antioxidant efficacy of *Allium* species. This knowledge can guide the development of more effective antioxidant agents based on *Allium*-derived compounds. The regular consumption of *Allium* vegetables, such as garlic and onions, can significantly decrease the risk of cancer in organs, including the brain, breast, esophagus, liver, lung, prostate, skin, and stomach.

*Allium* species, with their rich phytochemical content, offer a multifaceted approach to combating oxidative stress and cancer. The antioxidant effects are primarily due to the direct scavenging of free radicals, modulation of antioxidant enzymes, and activation of the Nrf2 pathway. The anticancer effects involve the induction of apoptosis, inhibition of cell proliferation, suppression of angiogenesis, and modulation of immune responses. These mechanisms highlight the potential of *Allium* phytochemicals as complementary therapies in the prevention and treatment of cancer.

The antioxidant and anticancer activities of *Allium* species indicate their potential against diseases driven by reactive oxygen species. Therefore, studies on *Allium* plants can lead to innovative strategies in oxidative and cancer treatment and prevention, offering new avenues for pharmaceutical exploration and dietary interventions, utilizing their remarkable potential as effective agents.

## Figures and Tables

**Figure 1 ijms-25-08079-f001:**
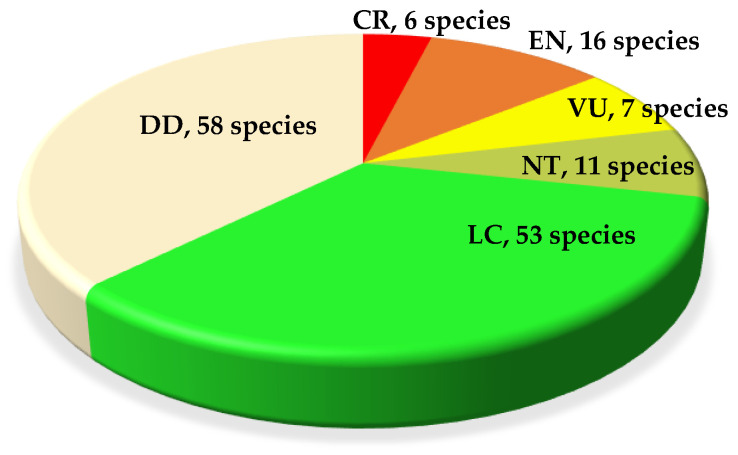
IUCN Red List category of *Allium* species, depicting the percentage of species classified as critically endangered (CR), endangered (EN), vulnerable (VU), near threatened (NT), least concern (LC), and data-deficient (DD).

**Figure 2 ijms-25-08079-f002:**
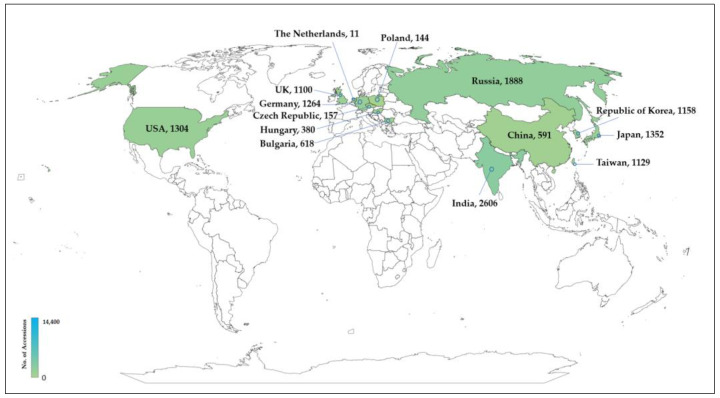
The major genebanks worldwide with *Allium* accessions.

**Figure 3 ijms-25-08079-f003:**
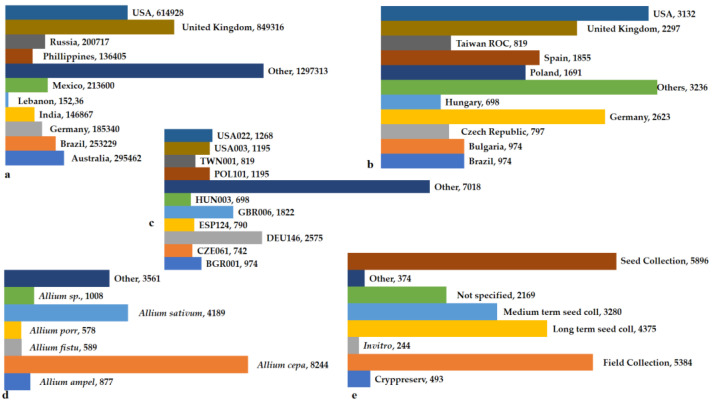
Countries with total crop (**a**), *Allium* (**b**) accessions, accession-holding institute (**c**), species name provided to accession during the submission (**d**), and germplasm storage type (**e**) from the data accessed through Genesys on 30 December 2023.

**Figure 4 ijms-25-08079-f004:**
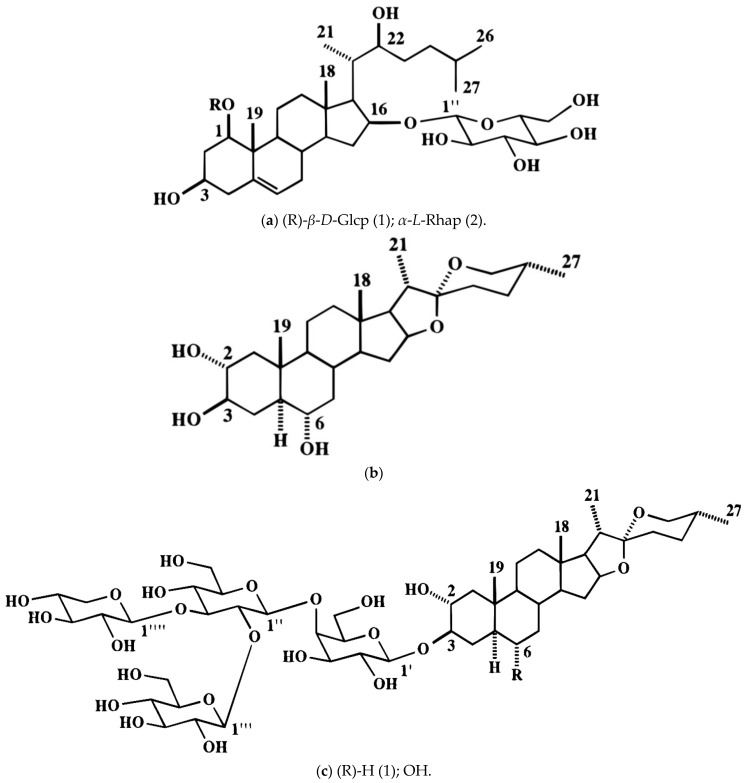
Steroidal glycosides from *A. jesdianum* Boiss & Buhse. (**a**) (22S)-cholest-5-ene-1*β*,3*β*,16*β*,22-tetrol 1,16-di-*O*-*β*-*d*-glucopyranoside (1), (22S)-cholest-5-ene-1*β*,3*β*,16*β*,22-tetrol 1-*O*-*α*-l-rhamnopyranosyl 16-*O*-*β*-*d*-glucopyranoside (2). (**b**) (25R)-5*α*-spirostane-2*α*,3*β*-diol 3-*O*-{*O*-*β*-*d*-glucopyranosyl-(1→2)-*O*-[*β*-*d*-xylopyranosyl-(1→3)]-*O*-*β*-*d*-glucopyranosyl-(1→4)-*β*-*d*-galactopyranoside}(F-gitonin). (**c**) (25R)-5*α*-spirostane-2*α*,3*β*,6*α*-triol 3-*O*-{*O*-*β*-*d*-glucopyranosyl-(1→2)-*O*-[*β*-*d*-xylopyranosyl-(1→3)]-*O*-*β*-*d*-glucopyranosyl-(1→4)-*β*-*d*-galactopyranoside}.

**Figure 5 ijms-25-08079-f005:**
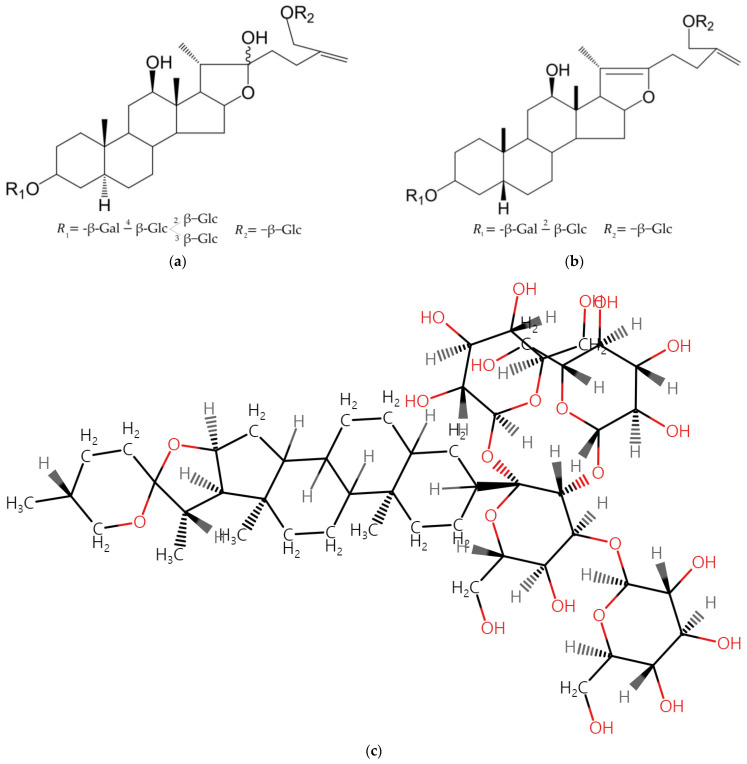
Steroidal saponins of *A. macrostemon* Bunge. (**a**) 26-*O*-*β*-*D*-glucopyranosyl-5*α*-furost-25 (27)-ene-3*β*, 12*β*, 22, 26-tetraol-3-*O*-*β*-*D*-glucopyranosyl (1→2) [*β*-D-glucopyranosyl (1→3)]-*β*-*D*-glucopyranosyl (1→4)-*β*-*D*-galactopyranoside. (**b**) 26-*O*-*β*-*D*-glucopyranosyl-5*β*-furost-20 (22)-25 (27)-dien-3*β*, 12*β*, 26-triol-3-*O*-*β*-*D*-glucopyranosyl (1→2)-*β*-*D*-galactopyranoside. (**c**) Macrostemonoside A of *A. macrostemon*.

**Table 1 ijms-25-08079-t001:** IUCN red-listed *Allium* species with high vulnerability.

*Allium* Species	Status and Year Assessed	Nativity
*A. akirense* N.Friesen & Fragman	CR (2015)	Israel
*A. iatrouanum* Trigas (un)	CR (2017)	Greece
*A. baytopiorum* Kollmann & Ozhatay	CR (2007)	E. Türkiye
*A. czelghauricum* Bordz.	CR (2007)	NE Türkiye
*A. corsicum* Jauzein, J.-M.Tison, Deschâtres & H.Couderc	CR (2010)	Corse
*A. marathasicum* Brullo, Pavone & Salmeri	CR (2016)	Cyprus
*A. pseudocalyptratum* Mouterde	EN (2016)	Lebanon, W. Saudi Arabia
*A. noeanum* Reut. ex Regel	EN (2019)	S.E. Türkiye, N. Syria, N. Iraq, N.W. and N. Iran
*A. sannineum* Gomb.	EN (2017)	Lebanon and Israel
*A. pseudoalbidum* N.Friesen & Özhatay	EN (2007)	Türkiye
*A. struzlianum* Ogan.	EN (2007)	S. Transcaucasus
*A. pervestitum* Klokov	EN (2011)	N.E. Black Sea Coast
*A. diomedeum* Brullo, Guglielmo, Pavone & Salmeri	EN (2015)	Italy
*A. garganicum* Brullo, Pavone, Salmeri & Terrasi	EN (2015)	S. Italy
*A. baeticum* Boiss.	EN (2018)	W. Central Portugal, S. Spain, NW. Africa
*A. agrigentinum* Brullo & Pavone	EN (2015)	Sicilia
*A. peroninianum* Azn.	EN (2016)	N. Türkiye
*A. therinanthum* C.Brullo, Brullo, Fragman, Giusso & Salmeri	EN (2015)	E. Mediterranean
*A. meronense* Fragman & R.M.Fritsch	EN (2016)	S. Lebanon to N. Israel
*A. basalticum* Fragman & R.M.Fritsch	EN (2016)	Lebanon to W. Jordan
*A. pycnotrichum* Trigas, Kalpoutz. & Constantin.	EN (2017)	Greece
*A. makrianum* C.Brullo, Brullo, Giusso & Salmeri	EN (2013)	E. Aegean Islands (Chios)
*A. schmitzii* Cout.	VU (2010)	E. Portugal to Central Spain
*A. pyrenaicum* Costa & Vayr.	VU (2010)	Pyrenees
*A. dumetorum* Feinbrun & Szel.	VU (2016)	Lebanon to Israel
*A. exaltatum* (Meikle) Brullo, Pavone, Salmeri & Venora	VU (2010)	Cyprus
*A. hemisphaericum* (Sommier) Brullo	VU (2018)	Sicilia
*A. castellanense* (Garbari, Miceli & Raimondo) Brullo, Guglielmo, Pavone & Salmeri	VU (2018)	Sicilia
*A. nebrodense* Guss.	VU (2015)	Sicilia
*A. pseudophanerantherum* Rech.f.	VU (2016)	Syria
*A. trichocnemis* J.Gay	VU (2018)	Algeria
*A. pelagicum* Brullo, Pavone & Salmeri	VU (2016)	Sicilia
*A. scaberrimum* J.Serres	VU (2018)	N.E. Spain, SE. France (Hautes Alpes) to Italy, N. Algeria to Tunisia
*A. lojaconoi* Brullo, Lanfr. & Pavone	NT (2011)	Malta
*A. altaicum* Pall.	NT (2013)	Siberia to N. China
*A. carmeli* Boiss.	NT (2014)	Syria to Israel
*A. libani* Boiss.	NT (2016)	Lebanon to S.W. Syria
*A. machmelianum* Post	NT (2018)	Syria
*A. calocephalum* Wendelbo	NT (2016)	S.E. Türkiye to N. Iraq
*A. melananthum* Coincy	NT (2010)	SE. Spain
*A. anzalonei* Brullo, Pavone & Salmeri	NT (2015)	Italy
*A. roylei* Stearn	NT (2013)	Afghanistan to W. Himalaya
*A. telmatum* Bogdanovic, Brullo, Giusso & Salmeri	NT (2015)	Croatia
*A. meikleanum* Brullo, Pavone & Salmeri	NT (2016)	Cyprus

CR—Critically endangered; EN—Endangered; VU—Vulnerable; NT—Near Threatened.

## Data Availability

Not applicable.

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
