# Peer review of "Alliums as Potential Antioxidants and Anticancer Agents"

_ijms, 2024, doi:10.3390/ijms25158079_

Round 1

Reviewer 1 Report

Comments and Suggestions for Authors

Dear authors,

The comments are highlighted in the PDF. Please, attention to use italic for species names and IUPAC rules of nomenclature.

A table of antioxidant activity should be included. And Table 2 should be improved. Conclusion should be clearer. 

Author Response

Dear reviewer,

Thank you for your valuable comments and suggestions to maximize our manuscript's quality. We have considered all the comments and addressed the issues raised as needed. For ease of follow-up, we have incorporated below point-by-point responses (in blue) for each of the comments raised.

Comments

 Line 42: Rewrite as “phenolic compounds, including flavonoids such as anthocyanins”

Response: Rewritten as mentioned

Line 67, 69, 112, 122, 124, 188, 260, 267, 272-283,

Response: Corrected.

Line 114: The methodology used for the literature review and inclusion criteria should be described and the table of antioxidant activities of Allium species

Response: methodology has been included in lines 73-83 and the Table of antioxidants activities of Allium species (Table 2) has been provided.

Line: Table 2: Allium species as anticancerous target improvisation 

Response: Anticancer activities of Allium species have been rectified and provided in Table 3.

Line 627-632: The conclusion part should be rewritten.

Response: The conclusion part has been rewritten and provided from lines 802-826.

Reviewer 2 Report

Comments and Suggestions for Authors

The document presents an interesting exercise in compiling information on the antioxidant and anticancer potential of the Allium genus. However, the authors need to thoroughly review how they organize and present the information beyond the introduction.

It is important to have greater clarity regarding the phytochemistry of the Allium genus and its relationship with different parts of the plant. This will provide the reader with better references to which part of the plant is used for culinary purposes or as a spice, compared to its medicinal uses and the bioactivities discussed. In this regard, the discussion is careless and overly general, making it difficult to adequately relate the most commonly used and most interesting parts of each species within the genus. The authors need to present a more detailed discussion on the families or groups of metabolites of interest, how they are distributed according to the part of the plant, and how they vary depending on important characteristics such as species and color. Including the chemical structures of some of the main groups of metabolites could be interesting.

The information is organized according to the antioxidant and anticancer activity discussed by species, but this organization is not very engaging to read. It does not include detailed information about the type of cancer, cell or in vivo models, and the proteins or biochemical systems targeted by molecules or preparations from the different species discussed. The authors should find a better way to integrate and present the information.

The words and phrases noted in line 241, Table 2 need to be reviewed. “Bioactive compound” is incorrect as it includes not only compounds but also extracts and other preparations. Additionally, biological activities that do not align with the purposes of the table should not be included. This also applies to other parts of the document, except for the introduction. The authors should focus on the antioxidant and anticancer activities. The quantities presented in lines 417 and 418 should be reviewed as they are incorrect. It is also recommended to expand and improve the conclusions.

Author Response

Dear reviewer,

Thank you for your valuable comments and suggestions to maximize our manuscript's quality. We have considered all the comments and addressed the issues raised as needed. For ease of follow-up, we have incorporated below point-by-point responses for each of the comments raised.

Comments

The document presents an interesting exercise in compiling information on the antioxidant and anticancer potential of the Allium genus. However, the authors need to thoroughly review how they organize and present the information beyond the introduction.

It is important to have greater clarity regarding the phytochemistry of the Allium genus and its relationship with different parts of the plant. This will provide the reader with better references to which part of the plant is used for culinary purposes or as a spice, compared to its medicinal uses and the bioactivities discussed. In this regard, the discussion is careless and overly general, making it difficult to adequately relate the most commonly used and most interesting parts of each species within the genus. The authors need to present a more detailed discussion on the families or groups of metabolites of interest, how they are distributed according to the part of the plant, and how they vary depending on important characteristics such as species and color. Including the chemical structures of some of the main groups of metabolites could be interesting.

The information is organized according to the antioxidant and anticancer activity discussed by species, but this organization is not very engaging to read. It does not include detailed information about the type of cancer, cell or in vivo models, and the proteins or biochemical systems targeted by molecules or preparations from the different species discussed. The authors should find a better way to integrate and present the information.

The words and phrases noted in line 241, Table 2 need to be reviewed. “Bioactive compound” is incorrect as it includes not only compounds but also extracts and other preparations. Additionally, biological activities that do not align with the purposes of the table should not be included. This also applies to other parts of the document, except for the introduction. The authors should focus on the antioxidant and anticancer activities. The quantities presented in lines 417 and 418 should be reviewed as they are incorrect. It is also recommended to expand and improve the conclusions.

Response: The Allium genus’s the main groups of metabolites has been discussed and provided in the lines from 143-182.

Phytochemistry of Allium 143

Allium species are renowned for their abundance of secondary metabolites with notable biological activities [24]. Economically significant Allium crops, such as garlic and onion, play a vital role in the daily diet, serving as both vegetables and medicinal ingredients. The genus Allium is a rich source of compounds with diverse bioactivities, both in vivo and in vitro. It contains a wide variety of metabolites. Both wild and cultivated species possess numerous key metabolites and other sulfur-rich substances, such as allicin [25]. Compounds derived from Allium species offer a wide range of health benefits, including antiviral, antibacterial, antifungal, antidiabetic, anticancer, antiplatelet, antispasmodic, antiseptic, antihelminthic, antithrombotic, antiasthmatic, carminative, antioxidant, anti-inflammatory, antihypertensive, hypoglycemic, hypotensive, lithontriptic, and hypo cholesterolemic properties [26]. Alkaloids, have been identified from this genus. Among these, three pyridine-N-oxide alkaloids with disulfide functional groups have been isolated from A. stiptatum: 2- (methyldithio) pyridine-N-oxide, 2-(methylthiomethyl)dithio] pyridine-N-oxide, and 2,2′-dithio-bis-pyridine-N-oxide. Additionally, the thiosulfinate natural product allicin, commonly found in the genus Allium, was synthesized from 2-(methyldithio) pyridine-N-oxide using a straightforward method by Kitson and Loomes and isolated as yellow and orange oils [25]. Flavonoids and their derivatives have been isolated from numerous Allium species, including A. cepa, A. sativum, A. schoenoprasum etc. These flavonoids exhibit antioxidant, anti-tumor, anti-inflammatory, and anti-mutagenic activities [27]. Various types of flavonoids, such as flavone, flavonol, and flavanone, have been identified across different Allium species. 166 Allium species contains the highest amounts of carbohydrates. Significant carbohydrate levels are also found in A. cepa and A.porrum. A. cepa contains fructo-oligosaccharides, while A. porrum contains active polysaccharides like glucuronic acid, galactose, and rhamnose. These polysaccharides, separated by water extraction, have high polyuronic and protein content but low sugar levels. A. sativum primarily contains reducing sugars such as glucose, fructose, sucrose, and maltose, which are known to stimulate the human immune system [28]. Sulfur compounds have been isolated, such as di-allyl sulfide, sulfinate, allylpropyl sulfide, and S-methyl-L-cysteine sulfoxide from Allium species. Allicin (diallyl thiosulfinate), a defense molecule from A. sativum, exhibits a broad range of biological  activities, including cancer, diabetes, and cardiovascular disease prevention [27]. Cysteine sulfoxides (alliin) have been isolated from several species, including A. sativum. Twenty- 178 seven volatile sulfur-rich compounds, including sulfides, disulfides, trisulfides, and tetrasulfides with ethyl, butyl, and pentyl groups, were extracted from A. tuberosum [27, 180 29]. Thioethyl and thiopentyl compounds have also been reported from A. schoenoprasum. These sulfur-containing compounds exhibit potent antioxidant and antitumor activities

Details on the antioxidant properties and importance has been provided in the lines from 183-210

Antioxidants play a crucial role in protecting our bodies from oxidative stress caused by free radicals and reactive oxygen species (ROS). This oxidative stress occurs due to an imbalance between the formation of ROS and their detoxification, leading to cellular damage. Chronic oxidative stress can result in several diseases, including cancer, coronary 187 heart disease, and osteoporosis [30]. Free radicals, such as superoxide anion (O2•), perhydroxyl radicals (HO2•), hydroxyl radicals (•OH), and nitric oxide, along with other 189 species like hydrogen peroxide (H2O2), singlet oxygen (1O2), hypochlorous acid (HOCl), 190 and peroxynitrite (ONOO-), can attack biomolecules, especially the polyunsaturated fatty acids in cell membranes [31]. The formation of ROS begins with the uptake of oxygen (O2), which activates NADPH oxidase, producing superoxide anion radicals. These radicals are then converted to hydrogen peroxide (H2O2) by superoxide dismutase (SOD) [32]. Antioxidants interrupt the chain reactions of free radicals by donating electrons to stabilize them without becoming free radicals themselves [33]. Antioxidants are classified into two types based on their activity: enzymatic and non- enzymatic. Enzymatic antioxidants include enzymes like glutathione peroxidase (GPx), catalase (CAT), and superoxide dismutase (SOD), which catalyze the neutralization of free radicals and ROS [34]. Non-enzymatic antioxidants, found in natural materials such as fruits and onions, include compounds like flavonoids, alkaloids, carotenoids, and phenolic groups [31, 34]. The antioxidant activity of these compounds can be assessed using various methods, including the DPPH free radical scavenging assay, oxygen radical absorbance capacity (ORAC) assay, trolox equivalent antioxidant capacity (TEAC) assay, ferric reducing antioxidant power (FRAP) assay, cupric reducing antioxidant capacity (CUPRAC) assay, and the reducing power assay. Allium species are significant for human health and comprise numerous health-beneficial bioactive compounds, i.e., flavonoids (polyphenols), sulfur compounds, vitamins, and minerals with antioxidant activity [35]. In this comprehensive review, we have extensively explored the antioxidant potential exhibited by diverse Allium species shown in table 2.

Response: lines 241, has been rectified and provided as “Invitro investigation of the antioxidant activity of (15%) hydroethanolic extracts from 328 different parts of endemic Italian Alliums” in lines 328-329

Response: Table 2, “Bioactive compound” is incorrect as it includes not only compounds but also extracts and other preparations. Has been rectified and provided as “table 3: Anticancer activities of different Allium species”.

Response: lines 417 and 418 has been corrected as “concentration of saponins, with 375 mg/g 507 while the bulb had 163.75 mg/g”.

Response: The conclusion part has been rewritten and provided from lines 802-826.

“Allium species are highly desirable and widely used in culinary arts including several wild species and used for traditional and folklore medicine somewhere known of regional curiosity, which were edible. Consumption of Allium reduces the chance of gastrointestinal tract cancer. Organosulfur compounds inhibit carcinogen activation enhance immune potential and have antioxidant, antibacterial, antifungal, antivirus, and anticancer properties. By analyzing the structure-activity relationships, we can understand the mechanism by which diverse bioactive compounds contribute to antioxidant efficacy of Allium species. This knowledge can guide the development of more effective antioxidant agents based on Allium-derived compounds. Regular consumption of Allium vegetables, such as garlic and onions, can significantly decrease the risk of cancerous diseases in organs like the brain, breast, esophagus, liver, lung, prostate, skin, stomach, etc. Allium species, with their rich phytochemical content, offer a multifaceted approach to combating oxidative stress and cancer. The antioxidant effects are primarily due to the direct scavenging of free radicals, modulation of antioxidant enzymes, and activation of the Nrf2 pathway. The anticancer effects involve the induction of apoptosis, inhibition of cell proliferation, suppression of angiogenesis, and modulation of immune responses. These mechanisms highlight the potential of Allium phytochemicals as complementary therapies in the prevention and treatment of cancer. The antioxidant and anticancer activity of Allium species is indicating their potential against diseases driven by reactive oxygen species. Therefore, the understanding the Allium plants will pave the way for innovative strategies in oxidative and cancer treatment and prevention, offering new avenues for pharmaceutical exploration and dietary interventions, emphasizing their remarkable potential as effective agents.”

Reviewer 3 Report

Comments and Suggestions for Authors

The manuscript explores the genus Allium, which includes plants such as onions, garlic, leeks, chives, and shallots, and their potential roles as antioxidants and anticancer agents. The study highlights the bioactive compounds present in these plants, such as allicin, flavonoids, and organosulfur compounds, which have demonstrated significant antioxidant and anticancer properties. However, the manuscript's content didn’t meet the scope of IJMS. The main comments are shown below:

1.  The authors should list more case studies or the main active components of each species, not just say flavonoids or organosulfur compounds.

2. The paper should include a mechanism part.

3. There is no Figure 1.

4. Figure 2 is tough to understand; what are the meanings of the abbreviations? Also, the percentage shown in the figure is different as described in the main text. Please add an abbreviation list at the end of the manuscript.

5. Please unify the format of the references.

Author Response

Dear reviewer,

Thank you for your valuable comments and suggestions to maximize our manuscript's quality. We have considered all the comments and addressed the issues raised as needed. For ease of follow-up, we have incorporated below point-by-point responses for each of the comments raised.

Comments

  1. The authors should list more case studies or the main active components of each species, not just say flavonoids or organosulfur compounds.

Response: The main active components of each species identified from the literatures used for this study has been mentioned in the Table 2 and 3.

  1. The paper should include a mechanism part.

Response: the discussion on the antioxidant and anticancer mechanism have been explained in the lines from 734 -801.

  1. There is no Figure 1.

Response: Rectified the typographical error

  1. Figure 2 is tough to understand; what are the meanings of the abbreviations? Also, the percentage shown in the figure is different as described in the main text. Please add an abbreviation list at the end of the manuscript.

Response: Figure 1, the IUCN redlist category of Allium species has been mentioned and the percentage mentioned in the script explained about the survival threats data provided in the IUCN database.

  1. Please unify the format of the references.

Response: The References are rectified in uniform manner.

Round 2

Reviewer 1 Report

Comments and Suggestions for Authors

Dear Authors,

Some of the requested corrections were made, however, there are aspects to be improved.

The resolution of chemical structures is not adequate. Including problems with bond angles.

Table 2 should be clear. It is relevant to describe the kind of extract used in the assay, eg. methanol extract, or the isolated compound tested.

There are some mistakes in IUPAC nomenclature which were pointed out in the previous version.

Please, attention to the attached file.

Comments on the Quality of English Language

Author Response

Dear Reviewer,

Thank you for your valuable comments and suggestions on our manuscript's quality. We have considered all the comments and addressed the issues raised as needed. For ease of follow-up, we have incorporated below point-by-point responses (in blue) for each of the comments raised.

Comments: The resolution of chemical structure in not adequate. Including problems with bond angles.

Response: The Chemical structure reported by the previous researcher from the Allium species has been provided with high resolution in Figure 4 and 5.

Comment: Table 2 should be clear. It is relevant to describe the kind of extract used in the assay, eg. Methanol extract, or the isolated compound tested.

Response: Table 2 and 3 has been updated with Extracts column (as per provided in literatures).

Comment: There are some mistakes in IUPAC nomenclature which were pointed out in the previous version.

Response: The Chemical nomenclatures names has been rectified as per reviewer mentioned.

Reviewer 2 Report

Comments and Suggestions for Authors

The manuscript has been improved with the incorporation of new and valuable information. However, the authors need to enhance the quality of the chemical structure figures, particularly Figure 5. It is essential to ensure better symmetry and higher image resolution. I recommend using chemical drawing software for this purpose and applying consistent settings or drawing properties to all figures.

Author Response

Dear Reviewer,

Thank you for your valuable comments and suggestions on our manuscript's quality. We have considered all the comments and addressed the issues raised as needed. For ease of follow-up, we have incorporated below point-by-point responses (in blue) for each of the comments raised.

Comments: The authors need to enhance the quality of the chemical structure figures, particularly Figure 5. It is essential to ensure better symmetry and higher image solution. I recommend using chemical drawing software for this purpose and applying consistent settings or drawing properties to all figures.

Response: The Chemical structure reported by the previous researcher from the Allium species has been provided with high resolution in Figure 4 and 5, that has been generated using the Marvin JS (https://marvinjs-demo.chemaxon.com/latest/).